# Quantification of the effect of hemodynamic occlusion in two-photon imaging of mouse cortex

**Baba Yogesh[1,2], Matthias Heindorf[1], Rebecca Jordan[3], Georg B Keller[1,2]\***

[1]Friedrich Miescher Institute for Biomedical Research, Basel, Switzerland; [2]Faculty of Natural Sciences, University of Basel, Basel, Switzerland; [3]Simons Initiative for the Developing Brain, University of Edinburgh, Edinburgh, United Kingdom

## eLife Assessment

This **important** study conducted experiments to quantify how neural activity independent changes in fluorescence might affect two-photon recordings when using diverse sensors. The researchers found a widespread presence of neural-activity-independent artifacts in two-photon imaging and provide **convincing** evidence that these artifacts are most likely caused by hemodynamic occlusion. Their findings underscore the importance of accounting for these artifacts when interpreting functional two-photon recordings.

**\*For correspondence:**
georg.keller@fmi.ch

**Competing interest:** The authors declare that no competing interests exist.

**Abstract** The last few years have seen an explosion in the number of tools available to measure neuronal activity using fluorescence imaging (Chen et al., 2013; Feng et al., 2019; Jing et al., 2019; Sun et al., 2018; Wan et al., 2021). When performed in vivo, these measurements are invariably contaminated by hemodynamic occlusion artifacts. In widefield calcium imaging, this problem is well recognized. For two-photon imaging, however, the effects of hemodynamic occlusion have only been sparsely characterized. Here, we perform a quantification of hemodynamic occlusion effects using measurements of fluorescence changes observed with GFP expression using both widefield and two-photon imaging in mouse cortex. We find that in many instances the magnitude of signal changes attributable to hemodynamic occlusion is comparable to that observed with activity sensors. Moreover, we find that hemodynamic occlusion effects were spatially heterogeneous, both over cortical regions and across cortical depth, and exhibited a complex relationship with behavior. Thus, hemodynamic occlusion is an important caveat to consider when analyzing and interpreting not just widefield but also two-photon imaging data.

## Introduction

Optical imaging of neuronal activity and neuromodulator concentration often involves recording the changes in fluorescence intensity of calcium or neuromodulator sensors expressed in neurons. In vivo, the hemodynamic changes in tissue around at the imaging location can modulate light transmission. Changes in blood volume through vasodilation and vasoconstriction as well as hemoglobin oxygenation change the transmission properties of the tissue. The combination of these effects is referred to as hemodynamic occlusion. Hemodynamic changes can be driven by local neuronal activity, a phenomenon known as neurovascular coupling (*Attwell et al., 2010*; *Iadecola and Nedergaard, 2007*) and are known to contribute significantly to activity measurements in techniques like fiber photometry (*Zhang et al., 2022*) and widefield microscopy (*Ma et al., 2016*; *Waters, 2020*). For these techniques, several methods have been proposed to control for the hemodynamic contribution to these

activity measurements (*Lohani et al., 2022*; *Valley et al., 2020*; *Zhang et al., 2022*). However, potential effects of hemodynamic signals on neuronal activity measurements acquired using two-photon imaging remain less well characterized. While it has been shown that arteriole dilation decreases fluorescence recorded in immediately underlying neurons, it is still unclear how large these effects are in a behaving animal where both neuronal activity dynamics and blood pressure and flow variance are likely larger than as previously measured under anesthesia (*Shen et al., 2012*).

Here, we characterized hemodynamic signals in two-photon imaging of mouse cortex during visuomotor behaviors. We expressed an activity independent marker (GFP) in mouse cortical neurons and measured fluorescence changes using two-photon imaging while mice were interacting with a virtual environment. We found changes in GFP fluorescence in response to locomotion and visual stimuli of a magnitude comparable to that measured with common activity sensors like GCaMP and GRAB variants. GFP signals were apparent both in population average responses and at individual neuron level. Moreover, these signals were heterogeneous over dorsal cortex and cortical layers. We compared the size of GFP signals to the calcium responses reported by GCaMP6 sensors (*Chen et al., 2013*), and to the neuromodulator signals reported by the GRAB sensors for dopamine (GRAB-DA1m), serotonin (GRAB-5HT1.0), acetylcholine (GRAB-ACh3.0), and norepinephrine (GRAB-NE1m). We found that a large fraction of the GRAB response to the different neuromodulators was explained by the GFP signal, consistent with an earlier study characterizing this relationship for GRAB-5HT1.0 and GRAB-5HT3.0 (*Ocana-Santero et al., 2024*). Based on our findings, we speculate that the primary driver of the hemodynamic signals in two-photon imaging is hemodynamic occlusion. If so, then this problem is likely primarily mitigated by using high dynamic range sensors with low baseline fluorescence, and best controlled for using activity-independent fluorescence measurements.

## Results

To quantify hemodynamic occlusion in two-photon microscopy, we imaged GFP in mouse cortex during behavior. We used an AAV vector (AAV2/1-Ef1α-eGFP-WPRE) injected in primary visual cortex (V1) and anterior cingulate cortex (ACC) to express GFP in cortical neurons (*Figure 1—figure supplement 1*). Given that the Ef1α promoter biases expression to excitatory neurons, we estimate that most (95%) of the GFP-expressing neurons were excitatory (*Attinger et al., 2017*). We then used a two-photon microscope to record GFP signals while mice explored a virtual tunnel. Mice were head-fixed on a spherical treadmill surrounded by a toroidal screen and exposed to a set of visuomotor conditions known to activate neurons in V1 and ACC (*Attinger et al., 2017*; *Keller et al., 2012*; *Leinweber et al., 2017*; *Zmarz and Keller, 2016*). First, mice were exposed to a closed loop condition during which running speed was coupled to visual flow speed in a virtual tunnel. Mice were also exposed to an open loop condition, during which locomotion and visual flow in the virtual tunnel were not coupled, to darkness, and to the presentation of full-field drifting gratings. Throughout all experimental conditions, mice were free to locomote on the spherical treadmill and did so voluntarily. Unless stated otherwise, we will use 'response' to refer to the observed change in GFP fluorescence.

### Locomotion onset and visual stimuli resulted in strong changes in GFP population responses

At the onset of locomotion, we found a transient increase in apparent GFP fluorescence in the population response of layer 2/3 (L2/3) neurons in V1 (*Figure 1A*). This increase in fluorescence of approximately 1% $\Delta F/F_0$ was only marginally smaller than the fluorescence changes typically observed at locomotion onset with genetically encoded calcium indicators (approximately 2% $\Delta F/F_0$ for GCaMP6f (*Widmer et al., 2022*; *Yogesh and Keller, 2023*), approximately 1.5% $\Delta F/F_0$ for GCaMP3 (*Keller et al., 2012*), etc.). By contrast, presentation of full-field drifting gratings resulted in a decrease of the GFP population signal (*Figure 1B*). We observed a similar decrease in GFP population signal on the presentation of an optogenetic stimulation light (637 nm, 10 mW after the objective) directed at V1 with either the virtual reality turned on or in the dark (*Figure 1C*). This response was likely visually driven as the decrease was twice as strong in dark than when the virtual reality was on, in line with the stimulation light being more visually salient in dark. Visuomotor mismatches (pauses in the visual flow feedback during locomotion), resulted in an average increase of apparent GFP fluorescence (*Figure 1D*). By design, visuomotor mismatch only occurs during times of locomotion.

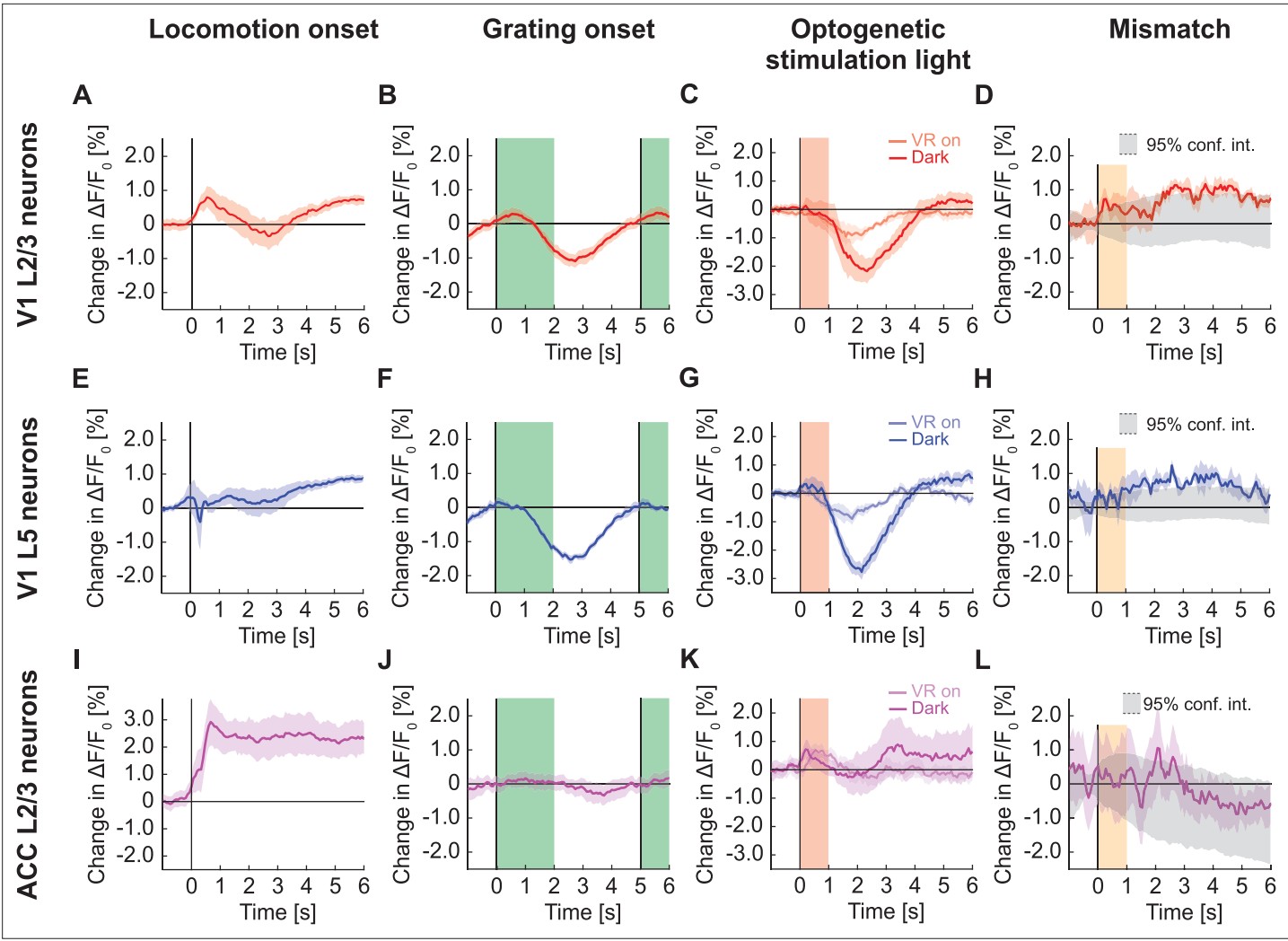

**Figure 1.** GFP responses in two-photon imaging of neurons in V1 and ACC. (**A**) Average change in apparent GFP fluorescence in L2/3 neurons in V1 on locomotion onset. Mean (solid lines) and the bootstrap SE (shading) are calculated as hierarchical bootstrap estimate for each time bin. (**B**) As in (**A**), but for grating onsets in L2/3 neurons in V1. Green shading marks the duration of the gratings. (**C**) As in (**A**), but for responses to optogenetic stimulation light in L2/3 neurons in V1, with VR on (light red) or in dark (dark red). Pink shading marks the duration of the light stimulus. (**D**) As in (**A**), but for visuomotor mismatch in L2/3 neurons in V1. Orange shading marks the duration of visuomotor mismatch. As mismatch events are defined to occur during times of locomotion, and locomotion itself drives GFP responses, we would expect to find an increase in these signals by chance at mismatch. To correct for this, we quantified the distribution of signal amplitude on random triggers during locomotion (95% confidence interval, gray shading). (**E**) As in (**A**), but for L5 neurons in V1. (**F**) As in (**B**), but for L5 neurons in V1. (**G**) As in (**C**), but for L5 neurons in V1. (**H**) As in (**D**), but for L5 neurons in V1. (**I**) As in (**A**), but for L2/3 neurons in ACC. (**J**) As in (**B**), but for L2/3 neurons in ACC. (**K**) As in (**C**), but for L2/3 neurons in ACC. (**L**) As in (**D**), but for L2/3 neurons in ACC.

The online version of this article includes the following figure supplement(s) for figure 1:

**Figure supplement 1.** Example imaging sites.

**Figure supplement 2.** GFP response heatmaps.

**Figure supplement 3.** GFP response in V1 L2/3 on locomotion onset in closed loop was not a linear sum of component responses.

Because apparent GFP fluorescence increases during locomotion, we estimated the increase expected by chance using random triggers during locomotion to compute the 95% confidence interval in *Figure 1D, H and L*. GFP responses to mismatch were only briefly outside these 95% confidence intervals. Thus, most stimuli tested drive measurable GFP responses at the population level. However, responses in individual neurons varied considerably (*Figure 1—figure supplement 2*).

To test whether stimulus triggered GFP responses depend on imaging depth and cortical area, we repeated these experiments in layer 5 (L5) of V1 and in L2/3 of ACC. In L5, we found that apparent

fluorescence changes at locomotion onset were smaller than those observed in L2/3 (*Figure 1E*). As in L2/3, full-field drifting grating stimuli (*Figure 1F*) and optogenetic stimulation light (*Figure 1G*) both resulted in a significant decrease in apparent GFP fluorescence, while visuomotor mismatch responses were only barely above chance (*Figure 1H*). In L2/3 of ACC, locomotion onset resulted in a strong increase of apparent GFP fluorescence (*Figure 1I*), while we found only small grating responses (*Figure 1J*) and a positive response to optogenetic stimulation light (*Figure 1K*). We again found no evidence of visuomotor mismatch responses (*Figure 1L*). Thus, apparent GFP fluorescence changes can vary both as a function of recording depth and cortical area.

The observation that average population responses of GFP and calcium indicators are similar does not mean that the dynamic range of individual neurons is similar. When comparing the distribution of peak $\Delta F/F_0$ responses, it is evident that peak responses are much larger for calcium indicators (*Figure 2A and B*). This is still the case when comparing trial averaged responses (*Figure 2C–E*), but the difference between GFP and calcium indicator response distributions is smaller, in particular for locomotion onsets (*Figure 2C*). The main driver for this effect is likely the highly variable stimulus triggered calcium responses of many neurons.

## Individual neurons showed significant GFP responses to different stimuli

Next, we tested whether GFP signals are strong enough to result in significant responses when looking at individual neurons. Raw GFP traces exhibited fluctuations reminiscent of calcium responses but tended to be of lower peak amplitude (*Figure 2A*). It is possible that small, correlated changes in GFP signal, when averaged over the population, result in a significant response indistinguishable from calcium signals reported with GCaMP but are negligible at the level of individual neurons. Thus, we quantified the fraction of neurons that are significantly responsive to different stimuli. GFP expression levels were high enough for all neurons in our dataset such that peak $\Delta F/F_0$ response amplitudes were independent of baseline $F_0$ (*Figure 3—figure supplement 1*). Commensurate with the population averages (*Figure 1*), we found a surprisingly large fraction of neurons in V1, in both L2/3 (*Figure 3A*) and L5 (*Figure 3B*) responsive to locomotion onset and grating stimuli. In both locations, the fraction of neurons responsive to visuomotor mismatch was not different from chance. In ACC L2/3 (*Figure 3C*), a significant fraction of GFP-labeled neurons was responsive to locomotion onset, while the fraction of neurons responsive to gratings and visuomotor mismatch was not different from chance. In all cases, the fraction of neurons responsive as measured with GFP was similar to the fraction of neurons responsive as measured with GCaMP (*Figure 3D–F*). Thus, even at the single neuron level, hemodynamic influence on GFP remains sizable.

## Changes in GFP fluorescence correlated with blood vessel dilation

One possible source of changes in GFP fluorescence is brain motion. However, we only find detectable brain motion during locomotion onsets and not the other stimuli (*Figure 4—figure supplement 1*), and even during locomotion onsets the movements are comparably small and follow different kinematics than the GFP signals. Finally, brain motion artefacts should on average decrease fluorescence, as they will move the cell out of the region of interest (ROI). Thus, we speculated that the dominant change in GFP fluorescence is driven by hemodynamic occlusion (see *Discussion* for alternative possibilities). Blood absorbs light. This absorption changes both as a function of blood volume and blood oxygenation. Thus, the blood vessels on the cortical surface that are between the GFP-labeled neurons and the objective of the microscope act to variably occlude the fluorescence measurement. As they constrict, dilate, or change oxygenation, they influence the transmission of both excitation and emission light similar, and thus act as a time varying occluder, similar to how a liquid crystal display (LCD) can be used to generate image sequences. One testable prediction of this hypothesis is that the cross-section of blood vessels visible in a given field of view should correlate inversely with the apparent GFP fluorescence of surrounding cells. This makes the simplifying assumption that the cross-section of surface blood vessels, which likely have the strongest contribution to occlusion, correlates with that of blood vessels in the imaging plane. To quantify the correlation between blood vessel cross-section and apparent GFP fluorescence, we first estimated the cross-section of selected blood vessels in the imaging plane (*Figure 4A–C*). Blood vessels in V1 dilated systematically, for example upon light stimulation (*Figure 4A1 and B1*), and contracted, for example on locomotion onset in ACC (*Figure 4C1*).

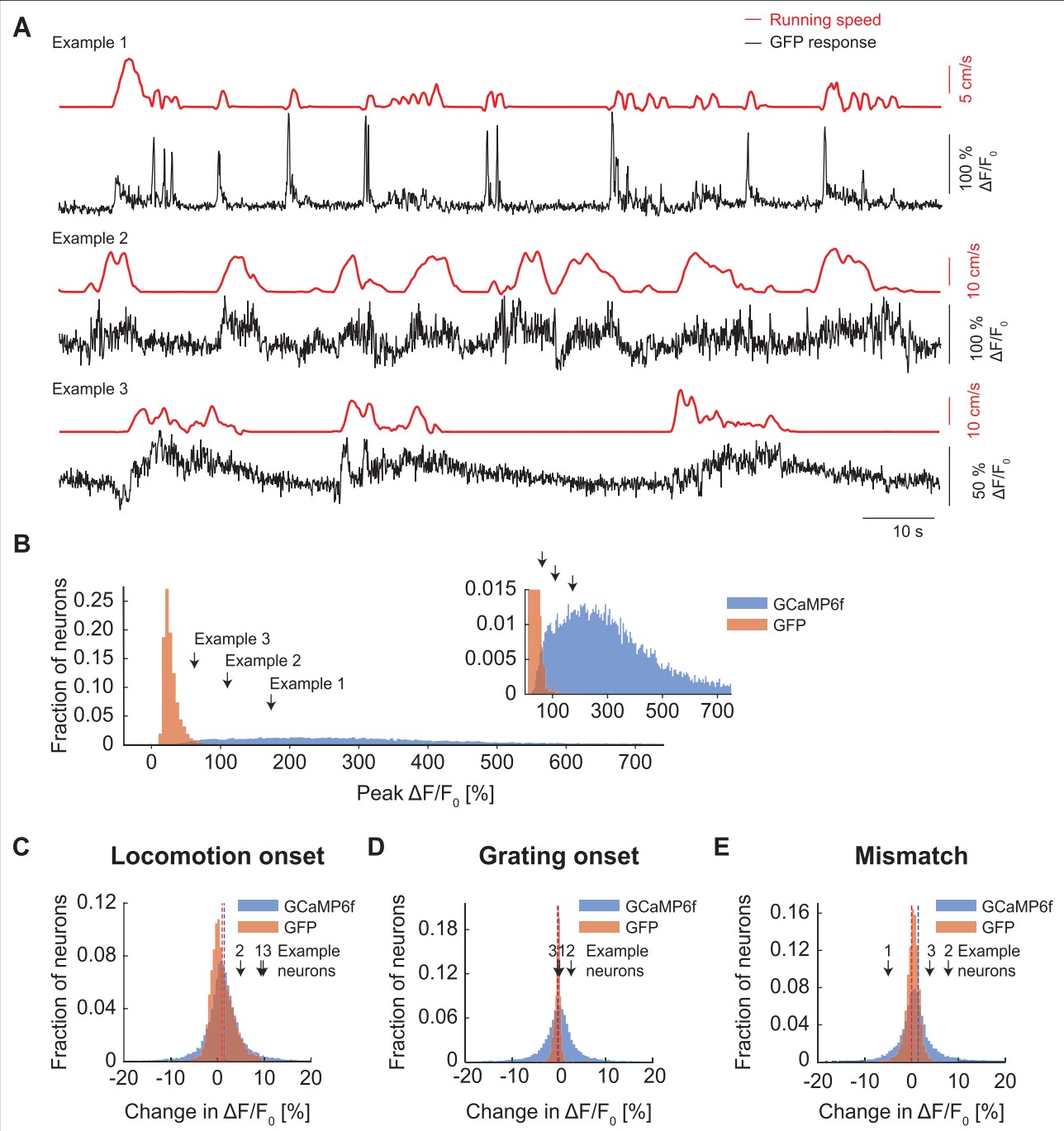

**Figure 2.** Peak responses of GCaMP6f were larger than those observed with GFP. (**A**) Example GFP response traces (black) of three neurons with corresponding running speed traces (red). (**B**) Peak responses of GCaMP6f (blue) were typically an order of magnitude larger than those observed with GFP (orange). Inset is a zoomed-in version of the same data. (**C**) Histogram of trial averaged responses on locomotion onset measured with two-photon imaging of calcium indicator (blue) and GFP (orange). Dotted lines indicate the mean response of the distribution of locomotion onset response in calcium indicator (blue) and GFP (orange). (**D**) As in (**C**), but for grating onset. (**E**) As in (**C**), but for visuomotor mismatch.

The estimation of blood vessel cross-section worked well for high-contrast boundaries but failed in many instances of low-contrast boundaries due to noise in the boundary estimates in single imaging frames. A proxy for blood vessel cross-section that does not rely on thresholding is the average fluorescence of a ROI of fixed size and fully contained within the blood vessel (*Figure 4A2, B2 and C2*). The intuition here is that there should be no fluorescence inside the blood vessel. Thus, fluorescence measured inside the blood vessel must come from the fact that the point spread function of the

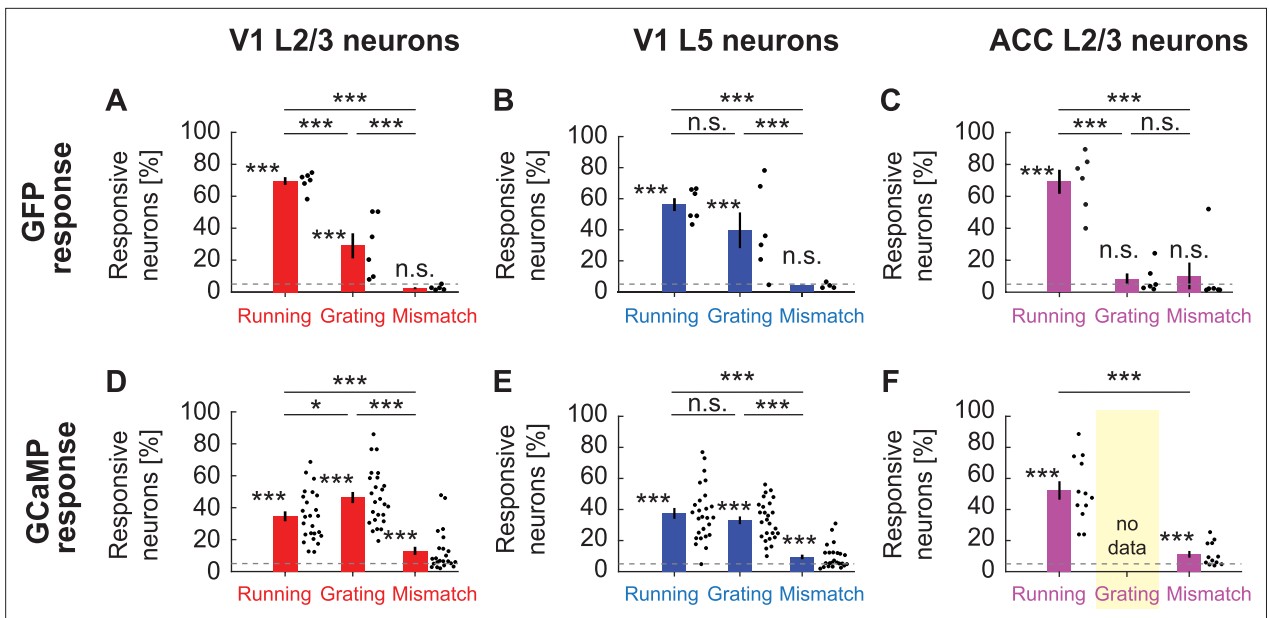

**Figure 3.** Fraction of neurons with significant GFP responses to different stimuli. (**A**) The fraction of GFP expressing V1 L2/3 neurons responsive to locomotion, full-field grating, and visuomotor mismatch onset, quantified for all imaging sites. Each dot is an imaging site, jittered along the x-axis to make them visible. Whiskers mark the SEM. Dashed line indicates chance levels (p=0.05). n.s.: not significant; *: p<0.05; **: p<0.01; ***: p<0.001; see *Supplementary file 1* for all statistical information. (**B**) As in (**A**), but for V1 L5 neurons. (**C**) As in (**A**), but for ACC L2/3 neurons. (**D**) As in (**A**), but for V1 L2/3 neurons expressing GCaMP6f. (**E**) As in (**A**), but for V1 L5 neurons expressing GCaMP6f. (**F**) As in (**A**), but for ACC L2/3 neurons expressing a GCaMP variant (data are from experiments using GCaMP3, GCaMP5, GCaMP6s, and GCaMP6f). Note, we have no grating presentation for this population.

The online version of this article includes the following figure supplement(s) for figure 3:

**Figure supplement 1.** Peak GFP response of neurons cannot be explained by GFP expression levels.

microscope extends to regions outside of the blood vessel. As the blood vessel contracts (or dilates) more (or less) of GFP in the tissue around the blood vessel will move into the point spread function. Comparing both metrics to the average apparent GFP fluorescence changes of the surrounding cells (*Figure 4A3, B3 and C3*), we found that both blood vessel area and fluorescence explain over 80% of the variance of apparent GFP fluorescence changes (*Figure 4A4, B and C*) in these example data. Here, variance explained is defined as the squared linear correlation coefficient between blood vessel area and fluorescence. Quantifying this for all data, we found that the fraction of variance explained was 50% or higher for most event triggered averages (*Figure 4D*) as well as across the entire recording duration (*Figure 4E* and *Figure 4—figure supplement 2*). The variance explained tended to be lower in ACC than in V1; we suspect that this is caused by the superior sagittal sinus contributing more strongly to hemodynamic occlusion in ACC, and that the diameters of local blood vessels are less well correlated with that of this large blood vessel. Consistent with vasoconstriction causing the increase of signal on locomotion onset, we sometimes find patterns of fluorescence changes that look like blood vessels that pass immediately above the imaging plan (*Figure 4—figure supplement 3A and B*). More typical however, is a broad increase that likely is driven by hemodynamic changes in blood vessels closer to the surface of the brain (*Figure 4—figure supplement 3C and D*).

## GFP responses were visuomotor context sensitive and could not be explained by a linear combination of responses to constituent stimuli

Next, we investigated how stereotypical GFP responses are across different visuomotor conditions and cortical regions, and whether GFP responses can be explained by a linear combination of a locomotion driven component and a visually driven component. We found that in V1 L2/3, locomotion onset responses were similar in closed loop and open loop, but different in the dark (*Figure 5A*). During a closed loop locomotion onset, there is concurrent onset of visual flow with the locomotion onset. During open loop and dark locomotion onsets, there is no concurrent onset of visual flow, and

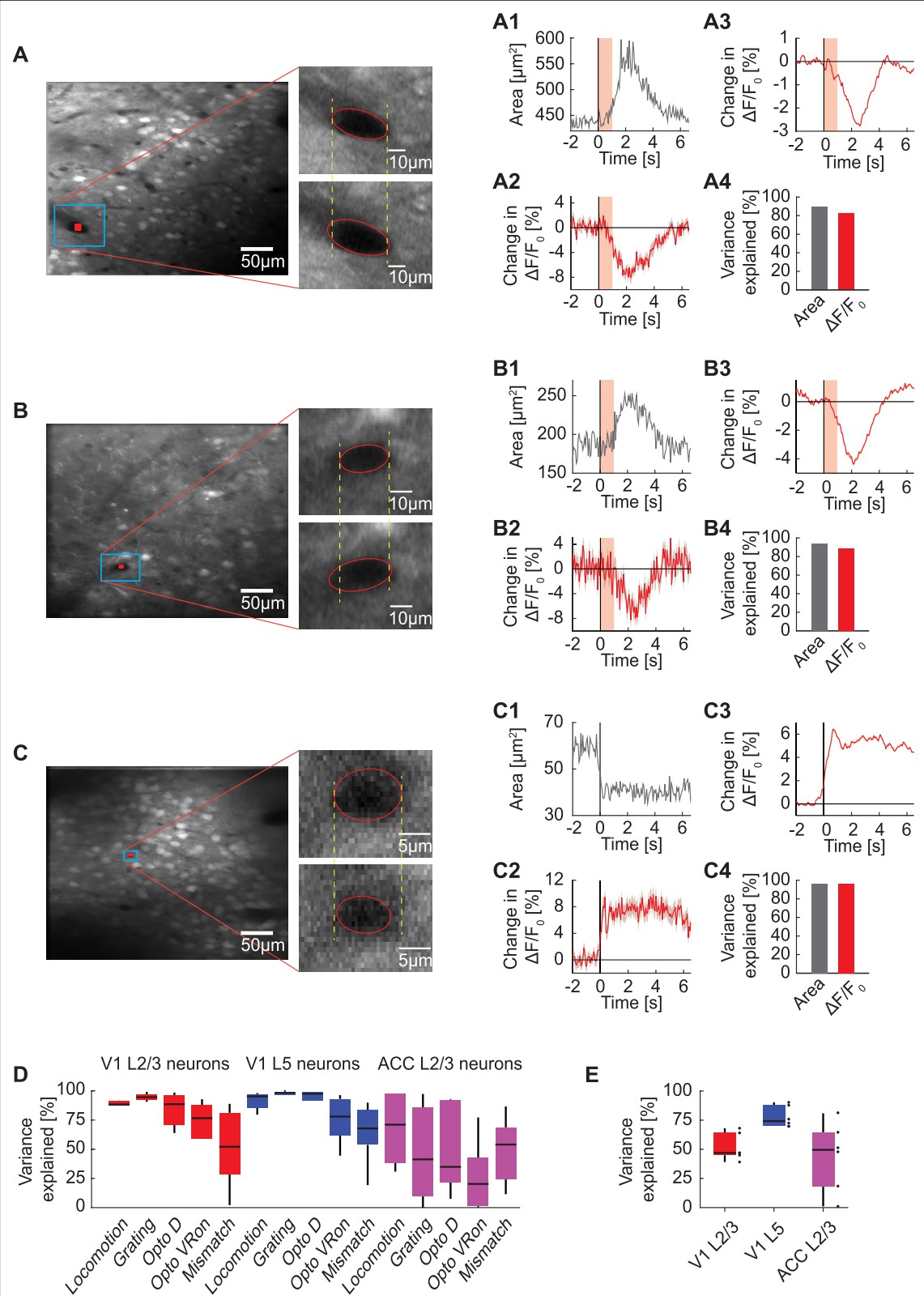

**Figure 4.** Correlation between blood vessel area and GFP responses. (**A**) Example average fluorescence image from an imaging site in L2/3 of V1 under a two-photon microscope. The fluorescence signal is averaged over 22 trials of optogenetic stimulation light directed at V1 while the mouse was in the dark. Shown on the right is an example blood vessel before (upper panel) and after (lower panel) stimulation. (**A1**) Average area of the example blood vessel shown in (**A**) (in blue box) on light stimulation. The area of the indicated blood vessel is extracted from the trial averaged two-photon image.

*Figure 4 continued*

Shading in pink indicates the optogenetic light presentation. (**A2**) Average change in GFP signal from a ROI positioned within the blood vessel (shown in **A** in red). Mean (red) and SEM (shading) were calculated over light stimulation trials. (**A3**) Average change in GFP signal from all neurons (268 neurons) simultaneously imaged in the same site. Mean (red) and SEM (shading) were calculated over light stimulation trials. (**A4**) Variance in average GFP signal in this example L2/3 site in V1 explained by area of the example blood vessel and by the change in GFP signal from a ROI within the blood vessel. (**B**) As in (**A**), from an imaging site in L5 of V1. (**C**) As in (**A**), from an imaging site in L2/3 of ACC on locomotion onset. (**D**) Percentage of variance in neuronal GFP signal to individual events explained by the change in GFP signal from blood vessel ROIs. Boxes show 25th and 75th percentile, central mark is the median, and the whiskers extend to the most extreme datapoints not considered outliers. (**E**) Percentage of variance in neuronal GFP signal over the entire recording explained by blood vessel GFP signal per imaging location. Each datapoint is one imaging site. Boxes show 25th and 75th percentile, central mark is the median, and the whiskers extend to the most extreme datapoints not considered outliers.

The online version of this article includes the following figure supplement(s) for figure 4:

**Figure supplement 1.** Brain motion estimate.

**Figure supplement 2.** The variance of neuronal GFP signal explained by blood vessel GFP signal as a function of smoothing window.

**Figure supplement 3.** Average locomotion onset response during two-photon GFP imaging.

they only differ in the average visual input. The similarity between open and closed loop running onsets could be explained by a dominant locomotion-related signal and a negligible visual onset response. If so, the smaller GFP responses in dark locomotion would need to be the consequence of the lower average visual input. This demonstrates that GFP responses, similar to neuronal calcium responses, can be non-linear combinations of component responses. This is perhaps not surprising, assuming local blood flow is primarily related to local neuronal activity. In L5 of V1, locomotion onset responses in open loop and dark conditions were comparable, while those in closed loop conditions had a delayed transient dip (*Figure 5B*). Thus, hemodynamic responses can exhibit a strong depth dependence. In ACC L2/3 neurons all three types of locomotion onsets resulted in increases of apparent fluorescence (*Figure 5C*). Thus, hemodynamic responses are difficult to predict from component signals (*Figure 1—figure supplement 3*), and do not generalize across depth or cortical areas. Finally, while there appears to be some correspondence to local neuronal activity in that locomotion onsets are more strongly modulated by visual context in V1 than in ACC, it is not immediately obvious how the GFP signals relate to responses measured with calcium indicators.

## Locomotion increased correlation in GFP signals

Another metric frequently used in the analysis of two-photon data are correlation-based analyses. Locomotion, for example, has been shown to decorrelate the activity of excitatory neurons in cortex (*Aydın et al., 2018*; *Dadarlat and Stryker, 2017*; *Eriksen et al., 2014*; *Yogesh and Keller, 2023*). Given the sizable fraction of neurons significantly responsive to locomotion, we speculated that correlations between neurons likely are influenced by hemodynamic responses. We tested this by quantifying

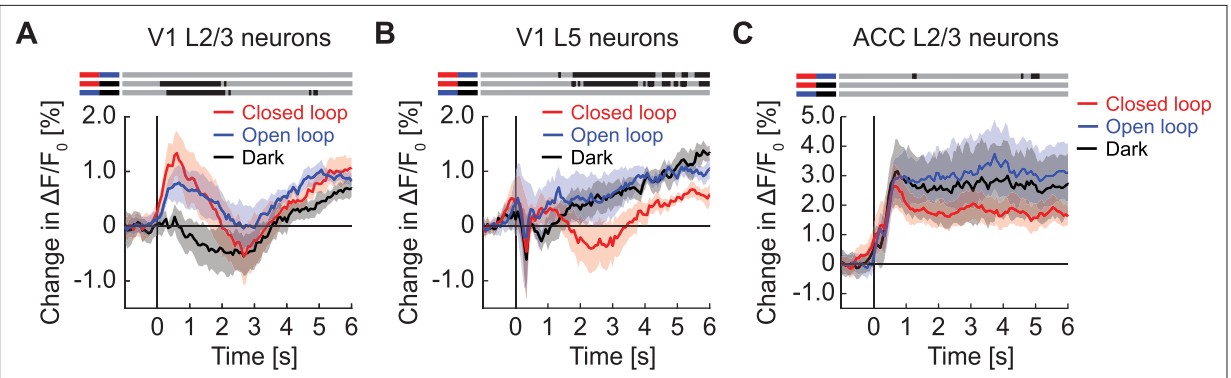

**Figure 5.** GFP responses at locomotion onset depended on visuomotor condition and cortical region. (**A**) Average GFP response in L2/3 neurons in V1 on locomotion onset in closed loop (in red), in open loop (in blue), and in darkness (in grey). Here and in subsequent panels, bins with a significant difference (p<0.01) are indicated by a black line above the plot (lines compared are marked on the left by pairs of colored horizontal bars); those with p>0.01 are marked gray. Mean (solid lines) and the bootstrap SE (shading) are calculated as hierarchical bootstrap estimate for each time bin. (**B**) As in (**A**), but for L5 neurons in V1. (**C**) As in (**A**), but for L2/3 neurons in ACC.

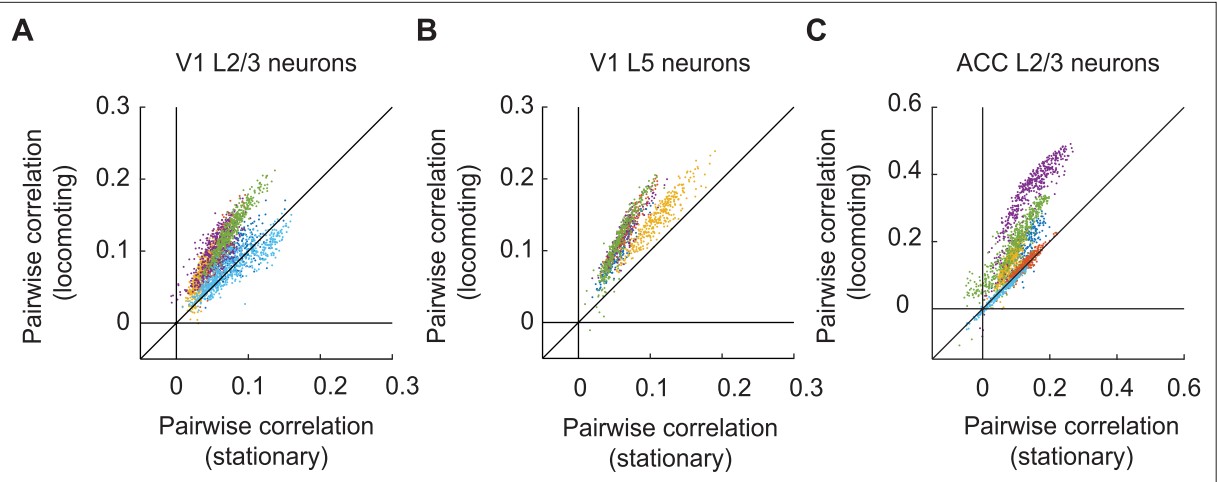

**Figure 6.** Locomotion increased pairwise correlation of GFP signals between neurons. (**A**) Average pairwise correlation in GFP signal of V1 L2/3 neurons while mice were stationary and while they were locomoting. Each dot is the mean pairwise correlation of one neuron to all other neurons in the same imaging site. Colors indicate data from different mice. (**B**) As in (**A**), but for V1 L5 neurons. (**C**) As in (**A**), but for ACC L2/3 neurons.

the effect of locomotion on the pairwise correlations of GFP signals between individual neurons. We found that pairwise correlations were systematically increased during locomotion (*Figure 6*). This was true for neurons in V1 L2/3 (*Figure 6A*), V1 L5 (*Figure 6B*), and ACC L2/3 (*Figure 6C*). Thus, hemodynamic responses can have a strong influence on pairwise correlations, and in the case of locomotion, this effect is opposite to that observed with calcium indicators (*Aydın et al., 2018*; *Dadarlat and Stryker, 2017*; *Erisken et al., 2014*; *Yogesh and Keller, 2023*). This is consistent with the interpretation that hemodynamic changes on locomotion occur at a larger spatial scale than changes in neuronal activity patterns and thus tend to increase correlations between neurons.

## GFP responses in widefield imaging were similar in magnitude to those in two-photon imaging

In widefield calcium imaging, hemodynamic responses are better characterized and widely recognized as a confounding problem (*Allen et al., 2017*; *Ma et al., 2016*; *Valley et al., 2020*). To directly contrast hemodynamic widefield signals with those observed in two-photon imaging, we repeated our experiments by performing widefield imaging of GFP responses. We expressed GFP pan-neuronally across dorsal cortex using either a retro-orbital injection of AAV-PHP.eB-EF1α-eGFP, or transgenic mice that express GFP under a *Fos* promoter (*Supplementary file 2*). As with two-photon imaging (*Figure 1*), and consistent with previous work (*Allen et al., 2017*; *Valley et al., 2020*), we found strong GFP responses across dorsal cortex (*Figure 7*). Locomotion and grating onsets resulted in GFP responses in both V1 (*Figure 7A and B*) and ACC (*Figure 7D and E*). During visuomotor mismatch, we found no significant response in either of these regions (*Figure 7C and F*). All these responses were greatly attenuated or absent when we imaged the cortex without expressing a fluorescent indicator (*Figure 7—figure supplement 1*), arguing in favor of hemodynamic occlusion of fluorescence from the genetically expressed indicators as the underlying cause for most of these responses in the GFP signal.

## GRAB sensors in cortex displayed similar responses to those recorded in GFP imaging

Finally, we compared the magnitude of hemodynamic responses to those recorded with the relatively low dynamic range GRAB sensors using both two-photon and widefield imaging. For this we used the GRAB-DA1m dopamine sensor (*Sun et al., 2018*), the GRAB-5HT1.0 serotonin sensor (*Wan et al., 2021*), the GRAB-ACh3.0 acetylcholine sensor (*Jing et al., 2019*), and the GRAB-NE1m (*Feng et al., 2019*) norepinephrine sensor (*Feng et al., 2019*). We first imaged GRAB-DA1m, GRAB-5HT1.0, and GRAB-ACh3.0 using two-photon imaging in L2/3 of V1. To estimate hemodynamic responses from GRAB sensors, we compared responses of ROIs selected in neuropil and those selected in blood

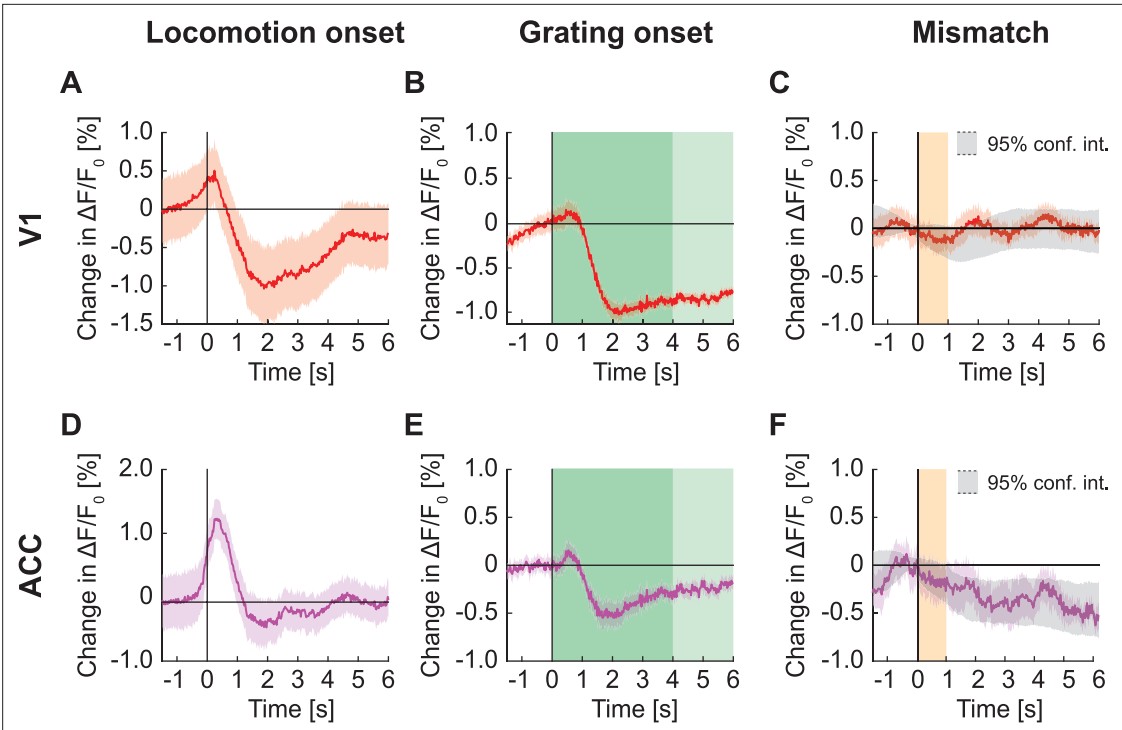

**Figure 7.** Signals in widefield fluorescence imaging of V1 and ACC in mice expressing GFP. (**A**) Average change in signal from V1 on locomotion onset. Mean (solid lines) and the bootstrap SE (shading) are calculated as hierarchical bootstrap estimate for each time bin. (**B**) As in (**A**), but for grating onset in V1. Grating duration were randomly selected between 4 s and 8 s (green shading). (**C**) As in (**A**), but on visuomotor mismatch in V1. Orange shading marks the duration of visuomotor mismatch. Gray shading marks the 95% confidence interval. (**D**) As in (**A**), but for change in signal from ACC on locomotion onset. (**E**) As in (**B**), but for change in signal from ACC on grating onset. (**F**) As in (**C**), but for change in signal from ACC on visuomotor mismatch.

The online version of this article includes the following figure supplement(s) for figure 7:

**Figure supplement 1.** Signals in widefield fluorescence imaging of V1 and ACC in mice not expressing a fluorophore.

vessels (*Figure 8A, E and I*). We reasoned that the signal in neuropil regions is a combination of a true GRAB signal and hemodynamic occlusion, while the signal in a ROI selected inside of a blood vessel should be dominated by the blood vessel diameter. We found that the GRAB-DA1m response to locomotion and full-field grating onsets (*Figure 8B and C*) was similar to GFP responses in V1 L2/3 (*Figure 1*). There were responses to locomotion and grating onset in the GRAB-DA1m signal (*Figure 8B and C*). In both cases neuropil and blood vessel ROI responses were similar. Also, similar to the GFP responses, we found no evidence of a response to visuomotor mismatch in the GRAB-DA1m signal (*Figure 8D*). We observed similar responses when imaging GRAB-5HT1.0 in V1 (*Figure 8F–H*). However, these responses were smaller in amplitude, likely driven by the overall dimmer fluorescence of the GRAB-5HT1.0 sensor. Thus, it is likely that both GRAB-DA1m and GRAB-5HT1.0 responses recorded with two-photon imaging are dominated by hemodynamic occlusion.

A response that was distinct from GFP, GRAB-DA1m, and GRAB-5HT1.0 signals was the response in GRAB-ACh3.0 on locomotion onset (*Figure 8J*). Here, we found a prominent increase in GRAB-ACh3.0 signal that was three-fold higher than either the GFP signal or the GRAB-ACh3.0 signals measured in blood vessel ROIs. This makes it more likely that the observed fluorescence changes are driven by a mixture of hemodynamic occlusion and GRAB-ACh3.0 responses. The GRAB-ACh3.0 response to grating onset (*Figure 8K*), however, was similar to GFP responses. And again, we found no response to visuomotor mismatch in GRAB-ACh3.0 (*Figure 8L*). To test whether the absence of response differences between GRAB-DA1m, GRAB-5HT1.0, and GFP signals was simply a result of imaging in V1, where both dopamine and serotonin release is weaker than in frontal areas of cortex (*Berger et al., 1991*; *Hamada et al., 2023*), we repeated these experiments in ACC. Also here, we found no evidence of GRAB responses that could not be explained by hemodynamic occlusion (*Figure 8—figure*

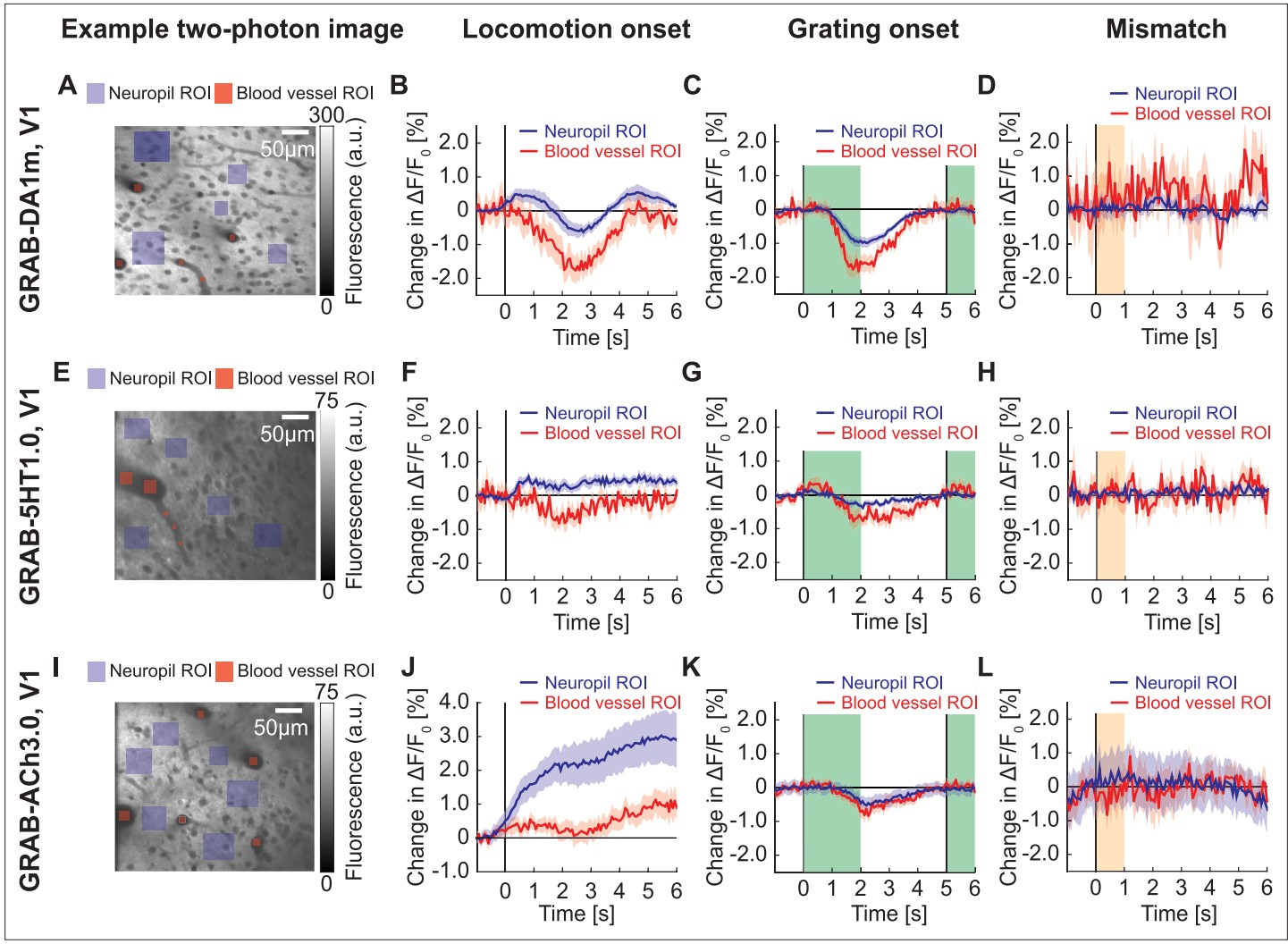

**Figure 8.** GRAB signals in V1 with two-photon imaging. (**A**) Example two-photon image taken in V1 of mouse expressing GRAB-DA1m, with neuropil ROIs (in violet) and blood vessel ROIs (in red). (**B**) Average change in GRAB-DA1m signal in V1 on locomotion onset in ROIs over neuropil (in black) and in ROIs in blood vessels (in red). Mean (solid lines) and the bootstrap SE (shading) are calculated as hierarchical bootstrap estimate for each time bin. (**C**) As in (**B**), but for response to grating onset. Green shading marks the duration of the gratings. (**D**) As in (**B**), but for response to visuomotor mismatch. Orange shading marks the duration of visuomotor mismatch. (**E**) Example two-photon image taken in V1 of mouse expressing GRAB-5HT1.0, with neuropil ROIs (in green) and blood vessel ROIs (in red). (**F**) Average change in GRAB-5HT1.0 signal in V1 on locomotion onset in ROIs over neuropils (in black) and in ROIs in blood vessels (in red). (**G**) As in (**F**), but for response to grating onset. (**H**) As in (**F**), but for response to visuomotor mismatch. (**I**) Example two-photon image of mouse expressing GRAB-ACh3.0 in V1, with neuropil ROIs (in green) and blood vessel ROIs (in red). (**J**) Average change in GRAB-ACh3.0 fluorescence in V1 in response to locomotion onset in ROIs over neuropil (in black) and in ROIs over blood vessels (in red). (**K**) As in (**J**), but for response to grating onset. (**L**) As in (**J**), but for response to visuomotor mismatch.

The online version of this article includes the following figure supplement(s) for figure 8:

**Figure supplement 1.** GRAB signals in ACC recorded with two-photon imaging.

*supplement 1*). A strong influence of hemodynamic signals was also apparent in widefield imaging of GRAB-NE1m. We found that the widefield GRAB-NE1m responses (*Figure 9*) were similar to GFP responses (*Figure 7*). Thus, in certain cases of low dynamic range sensors, one can readily measure responses, but these are not always easily distinguished from those driven by hemodynamic occlusion.

## Discussion

Our data demonstrate that in vivo two-photon fluorescence imaging during behavior is subject to relatively large and fast changes in measured fluorescence independent of any sensor typically used

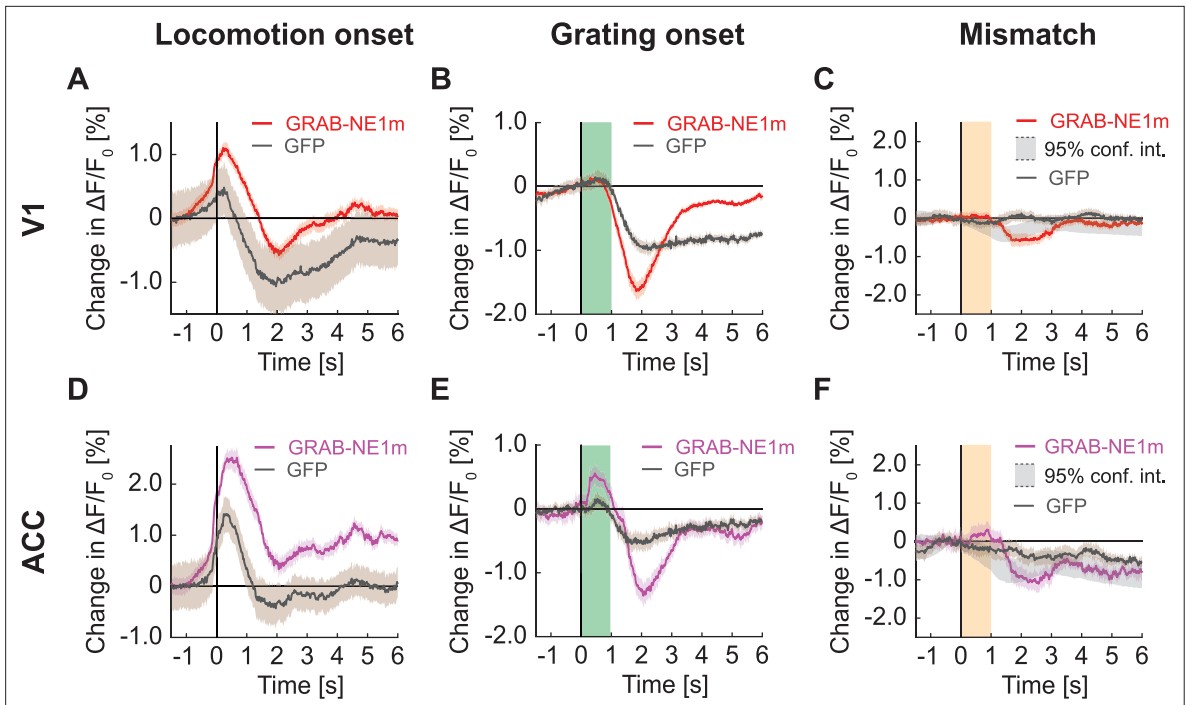

**Figure 9.** Signals of V1 and ACC in widefield imaging in mice expressing GRAB-NE1m. (**A**) Average change in signal from V1 on locomotion onset. Mean (solid lines) and the bootstrap SE (shading) are calculated as hierarchical bootstrap estimate for each time bin. Here and below, GFP response (in gray) from *Figure 7* overlaid. (**B**) As in (**A**), but for grating onset in V1. Green shading marks the duration of the gratings. (**C**) As in (**A**), but for visuomotor mismatch in V1. Orange shading marks the duration of visuomotor mismatch. Gray shading marks the 95% confidence interval. (**D**) As in (**A**), but for change in signal from ACC on locomotion onset. (**E**) As in (**B**), but for change in signal from ACC on grating onset. (**F**) As in (**C**), but for change in signal from ACC on visuomotor mismatch.

to measure functional calcium, voltage, or peptide changes. The surprising finding to us, was not that these changes are observed, but rather how large they are on a population average compared to fluorescence changes measured with functional sensors. There are several possible sources of these fluorescence changes:

## Brain movement artifacts

Many body movements result in the movement of the brain relative to the skull. In our experience, locomotion onset as well chewing and licking (or other movements that involve the masseter muscles) are the strongest drivers of such movements. Use of a cranial window prevents much of the movement in the axis perpendicular to the window (which typically is the optical axis). To mitigate this problem, we discard any experimental preparation with visible z-motion, and correct x/y motion using a full-frame registration algorithm. For locomotion onsets, brain movement kinematics are different in shape than the GFP responses we observe, and in the case of visual stimulation, we find no evidence of stimulus driven brain movements (*Figure 4—figure supplement 1*). Finally, given that the ROIs used to quantify fluorescence changes are selected based on a mean position of the neuron, movement away from the mean position should result in an average decrease in observed fluorescence. Thus, while there certainly are brain-movement-related artifacts that can strongly influence two-photon measurements if not properly mitigated, these artifacts cannot explain visual-stimulation-induced fluorescence changes, and likely only explain the initial, fast decrease of fluorescence observed in a subset of neurons at locomotion onset (*Figure 1—figure supplement 2*).

## Stimulation light contamination

Light from the visual display system can leak into the optical pathway of the two-photon microscope. This results in stimulus locked changes in fluorescence that can occur both for visual stimulation, but also for closed loop locomotion onsets, as locomotion triggers an onset of visual flow. This can be

partially mitigated by physically shielding the imaging site from stimulation light. However, complete shielding is difficult (e.g. some of the stimulation light likely travels through the eye to the brain and into the microscope – the fact that light quite efficiently travels this route is what has enabled convenient eye tracking in two photon imaging as the stimulation light from the two-photon laser illuminates the pupil from the inside). In two photon imaging, this problem can however be almost completely solved by synchronizing all light sources in the experimental setup to the dead times of the microscope (e.g. the turnaround times of the resonant scanner) (*Leinweber et al., 2014*), as we have also done here. Finally, stimulation light contamination exhibits essentially immediate onset and offset kinetics and is relatively simple to recognize. Thus, we are confident that stimulation light contamination does not significantly contribute to the effects we describe here.

## GFP's sensitivity to pH changes

GFP changes its fluorescence with changes in pH (*Kneen et al., 1998*). And intracellular pH systematically changes with neuronal activity. While the relationship between neuronal activity and intracellular pH is complex and not fully understood, increases in neuronal activity tend to result in decreases of intracellular pH. Thus, the activity-driven pH change should decrease GFP fluorescence. What we find, however, in many cases is a positive correlation between neuronal activity and GFP fluorescence. More importantly, however, intracellular pH changes driven by neuronal activity are on the order of 0.1 around a resting pH somewhere between 7 and 8 (*Chesler, 2003*; *Raimondo et al., 2013*). While GFP is indeed sensitive to pH, this is primarily the case for pH values below 6. The fluorescence changes above a pH of 7 are comparably small, and likely negligible when the changes are in the range of 0.1 (*Dos Santos et al., 2020*; *Kneen et al., 1998*). Thus, we do not think the pH sensitivity of GFP substantially contributes to the effects we observe.

## Hemodynamic tissue deformation

A second type of movement artifact is produced by local tissue deformations caused by blood vessel diameter changes. Cells in close proximity to a blood vessel are displaced when blood vessels dilate and contract. These effects are primarily visible in cells in the immediate vicinity (a few cell body diameters) of blood vessels. As with brain-movement-driven artefacts, the net effect of such tissue deformations is to displace the cell away from the ROI used to measure fluorescence and hence result they tend to result in apparent fluorescence decreases on average. These effects certainly also contribute to the signals we report but given the observed increase in fluorescence on running onset, the effect of hemodynamic tissue deformation are likely smaller compared to that of occlusion. In a subset of neurons, such deformations can of course also result in fast and large amplitude increases in fluorescence if a neuron is above or below the imaging plane at rest and moved into the imaging plane by the blood vessel. We suspect that for example the first example in *Figure 2A* is driven by such an effect. However, such high amplitude events are rare (*Figure 2B*). Moreover, these effects are primarily visible when there is a high fluorescence inside a ROI and low fluorescence immediately surrounding it (e.g. as is the case when expressing GFP or GCaMP). When expressing GRAB sensors, labeling is much more homogeneous, and there are no distinct contrast changes at the edges of ROIs. Thus, the fact that the hemodynamic-driven signals look very similar when labeling is primarily confined to somata (*Figure 1*), compared to using GRAB sensors (*Figure 8*), is consistent with our interpretation that hemodynamic tissue deformations are not the primary driver of the signals we report.

## Hemodynamic occlusion

Blood absorbs stimulation and emission light. As a function of volume and composition of blood, light absorption can change dynamically. Changes in neuronal activity in the cortex are associated with hemodynamic changes through neurovascular coupling (*Ruff et al., 2024*) that are layer specific (*Mächler et al., 2021*). Cells directly under arterioles have been shown to be occluded by changes in arteriole diameter (*Shen et al., 2012*). In brain structures like cortex that have a relatively dense blood vessel network on the surface, this problem is augmented as any blood vessel within the imaging light cone between the surface of the brain and the recorded cell will contribute to the occlusion. Using an objective with a numerical aperture of 1, this is a light cone with a half angle of approximately 45° . Thus, when for example imaging a neuron at a depth of 500 µm, all surface vessels in circle around location of the neuron of 1 mm diameter will contribute to the occlusion. Even ignoring occlusion

contributions that arise from changes to the composition of blood (e.g. oxygenation changes), the changes driven by blood vessel cross-section changes are likely sufficient to explain the effect sizes we observe. Thus, we conclude that hemodynamic occlusion is the primary contributor to the effects we observe. Some of the other factors discussed above certainly also contribute to the effects we describe, and there may yet be other influences we have not thought of yet. Importantly, however, all these factors similarly would contribute to signals measured with functional indicators.

### How to mitigate the problem

There are likely two approaches: use a second imaging channel for concurrent isosbestic illumination of the sensor or estimate the hemodynamic contribution in a separate set of experiments using GFP. For the calcium indicator GCaMP, isosbestic imaging has been shown to work in widefield imaging (*Allen et al., 2017*; *Couto et al., 2021*). However, imaging at the isosbestic wavelength concurrently with the functional measurement would require a second tunable two-photon light source. In the case of fiber photometry (*Zhang et al., 2022*) and widefield imaging (*Ma et al., 2016*; *Scott et al., 2018*; *Valley et al., 2020*,) hemodynamic influence is well documented, and there are correction methods published (*Allen et al., 2017*; *Ma et al., 2016*). But even here methods based on isosbestic illumination can be difficult to implement accurately, either because light sources used for isosbestic illumination are not tunable or not available at the exact wavelength necessary, or when using LEDs, have emission spectra are too broad. One could consider trying a model-based approach to estimate hemodynamic occlusion effects based on GFP measurements, as can be done for widefield imaging (*Valley et al., 2020*). However, because hemodynamic occlusion changes across the cortical surface and depth, separate GFP calibrations are likely necessary for different regions and layers imaged. Finally, a simple but laborious approach is to repeat key experiments using GFP to test how much of the functional effects can be explained by non-functional effects. While arguably more than a decade too late (*Keller et al., 2012*), this is what our aim was here.

Finally, given that hemodynamic changes are likely driven by neuronal activity through neurovascular coupling (*Ruff et al., 2024*), which can be highly specific to cortical layer (*Mächler et al., 2021*), these signals could be a useful proxy to estimate neuronal activity. Assuming one would identify a neuronal cell-type with activity that correlates well with local hemodynamic changes, blood vessel diameter measurements could be used as a relatively simple secondary measurement in functional two-photon imaging.

## Materials and methods

**Key resources table**

| Reagent type (species) or resource | Designation | Source or reference | Identifiers | Additional information |
|---|---|---|---|---|
| Strain, strain background (adeno-associated virus) | AAV2/1-Ef1α-eGFP-WPRE ($10^{12}$ GC/ml) | FMI vector core | vector.fmi.ch | |
| Strain, strain background (adeno-associated virus) | AAV2/1-Ef1α-GCaMP6f-WPRE ($10^{11}$–$10^{14}$ GC/ml) | FMI vector core | vector.fmi.ch | |
| Strain, strain background (adeno-associated virus) | AAV2/9-hSyn-GRAB-DA1m ($10^{12}$ GC/ml) | FMI vector core | vector.fmi.ch | |
| Strain, strain background (adeno-associated virus) | AAV2/9-hSyn-GRAB-5HT1.0 ($10^{13}$ GC/ml) | FMI vector core | vector.fmi.ch | |
| Strain, strain background (adeno-associated virus) | AAV2/9-hSyn-GRAB-ACh3.0 ($10^{13}$ GC/ml) | FMI vector core | vector.fmi.ch | |
| Strain, strain background (adeno-associated virus) | AAV-PHP.eB-EF1α-eGFP ($10^{13}$ GC/ml) | FMI vector core | vector.fmi.ch | |
| Strain, strain background (adeno-associated virus) | AAV-PHP.eB-hSyn-GRAB-NE1m ($10^{13}$ GC/ml) | FMI vector core | vector.fmi.ch | |
| Chemical compound, drug | Fentanyl citrate | Actavis | CAS 990-73-8 | Anesthetic compound |

*Continued on next page*

*Continued*

| Reagent type (species) or resource | Designation | Source or reference | Identifiers | Additional information |
|---|---|---|---|---|
| Chemical compound, drug | Midazolam (Dormicum) | Roche | CAS 59467-96-8 | Anesthetic compound |
| Chemical compound, drug | Medetomidine (Domitor) | Orion Pharma | CAS 86347-14-0 | Anesthetic compound |
| Chemical compound, drug | Ropivacaine | Presenius Kabi | CAS 132112-35-7 | Analgesic compound |
| Chemical compound, drug | Lidocaine | Bichsel | CAS 137-58-6 | Analgesic compound |
| Chemical compound, drug | Buprenorphine | Reckitt Benckiser Healthcare | CAS 52485-79-7 | Analgesic compound |
| Chemical compound, drug | Humigel | Virbac | - | Ophthalmic gel |
| Chemical compound, drug | Flumazenil (Anexate) | Roche | CAS 78755-81-4 | Anesthetic antagonist |
| Chemical compound, drug | Atipamezole (Antisedan) | Orion Pharma | CAS 104054-27-5 | Anesthetic antagonist |
| Chemical compound, drug | Metacam | Boehringer Ingelheim | CAS 71125-39-8 | Analgesic compound |
| Chemical compound, drug | N-Butyl-2-cyanoacrylate | Braun | CAS 6606-65-1 | Histoacryl |
| Chemical compound, drug | Dental cement (Paladur) | Heraeus Kulzer | CAS 9066-86-8 | |
| Genetic reagent (*Mus musculus*) | C57BL/6 | Charles River | - | |
| Genetic reagent (*M. musculus*) | B6J.129S6-Chat$^{tm2(Cre)Lowl}$/MwarJ Alias used here: ChAT-IRES-Cre | Jackson Laboratories | RRID:IMSR_JAX:028861 | Cre expression in cholinergic neurons |
| Genetic reagent (*M. musculus*) | B6J.FVB(Cg)- Tg(Tlx3-cre)PL56Gsat/Mmucd Alias used here: Tlx3-Cre | MMRRC | RRID:MMRRC_041158-UCD | Cre expression in a subset of L5 neurons |
| Genetic reagent (*M. musculus*) | B6.Cg- Tg(Fos/EGFP)1-3Brth/J Alias used here: fosGFP | Jackson Laboratories | RRID:IMSR_JAX:014135 | Cre expression in a subset of cortical neurons |
| Software, algorithm | MATLAB (2023b) | The MathWorks | RRID:SCR_001622 | Data analysis |
| Software, algorithm | LabVIEW | National Instruments | RRID:SCR_014325 | Hardware control |
| Software, algorithm | Two-photon acquisition software | Keller laboratory | sourceforge.net/p/iris-scanning/ | Data acquisition |
| Software, algorithm | Image data processing software | Keller laboratory | sourceforge.net/p/iris-scanning/calliope | Data processing |
| Software, algorithm | Python | python.org | RRID:SCR_008394 | Virtual reality |
| Software, algorithm | Panda3D | panda3d.org | RRID:SCR_021216 | Virtual reality |
| Other | Virtual reality and two-photon setup | *Leinweber et al., 2014*; *Leinweber et al., 2017* | N/A | Hardware setup |
| Other | OBIS 673 nm LX | Coherent | Cat#1187194 | Optogenetic stimulation laser |
| Other | LED | Prizmatix | UHP-T-595 | Sham stimulation |
| Other | Titanium headplate | FMI/ETHZ workshop | N/A | Mice head-fixation |
| Other | Dental drill | Meisinger | N/A | For craniotomy |

## Mice

We used a total of 35 C57BL/6 mice, 4 fosGFP mice, 6 Tlx3-Cre mice, and 23 ChAT-IRES-Cre mice. Both male and female mice, 6–16 weeks old at the start of the experiment, were used. See *Supplementary file 2* for details of mouse inclusion for different figures. Between experiments, mice were group-housed in a vivarium (light/dark cycle: 12/12 hr). All animal procedures were approved by and carried out in accordance with the guidelines laid by the Veterinary Department of the Canton of Basel-Stadt, Switzerland.

## Surgery

For all surgical procedures, mice were anesthetized with a mixture of fentanyl (0.05 mg/kg; Actavis), midazolam (5.0 mg/kg; Dormicum, Roche), and medetomidine (0.5 mg/kg; Domitor, Orion) injected intraperitoneally. Analgesics were applied perioperatively (2% lidocaine gel, meloxicam 5 mg/kg) and postoperatively (buprenorphine 0.1 mg/kg, meloxicam 5 mg/kg). Eyes were covered with ophthalmic gel (Virbac Schweiz AG). Depending on the experiment, we either implanted a cranial window to perform two-photon imaging or used crystal skull (or clear skull preparation in case of GRAB-NE1m imaging) for widefield imaging. Cranial windows were implanted over V1 and ACC as previously described (*Keller et al., 2012*; *Leinweber et al., 2014*). Briefly, using a dental drill, a 4 mm craniotomy was made over the right V1, centered 2.5 mm lateral and 0.5 mm anterior to lambda. A second craniotomy was made over right ACC, centered at midline, 0.5 mm anterior to bregma. After injection of an AAV vector carrying the reporter (see *Supplementary file 2* for details of virus), the exposed cortex was sealed with a 3 mm or 4 mm circular glass coverslip and glued in place using gel superglue (Ultra Gel, Pattex). The remaining exposed surface of the skull was covered with Histoacryl (B. Braun), and a titanium head bar was fixed to the skull using dental cement (Paladur, Heraeus Kulzer). For widefield experiments, we injected an AAV vector with PHP.eB capsid retro-orbitally (6 µl per eye of at least $10^{13}$ GC/ml) to drive expression throughout cortex, or imaged GFP expressed in a fosGFP mice, or imaged without any fluorophore. For crystal skull surgery, we surgically removed the skull plate overlying the dorsal cortex and superglued a crystal skull coverslip over the craniotomy. Prior to removing the skull plate overlying dorsal cortex, the location of bregma relative to other landmarks on the skull was recorded. For clear skull preparation, the skull was cleared with a three-component polymer (C&B Metabond, Parkell), and the crystal skull coverslip was directly attached to the skull. An epifluorescence image was taken to mark reference points on cortical surface. As with 4 mm cranial window implantation, a titanium head bar was fixed to the skull using dental cement. After surgery, anesthesia was antagonized by a mixture of flumazenil (0.5 mg/kg; Anexate, Roche) and atipamezole (2.5 mg/kg; Antisedan, Orion Pharma) injected intraperitoneally. Imaging commenced earliest 2 weeks after head bar implantation or 3 weeks after retro-orbital AAV injection.

## Virtual reality environment

The virtual reality setup is based on the design of Dombeck and colleagues (*Dombeck et al., 2007*). Briefly, mice were head-fixed and free to run on an air-supported spherical treadmill. The rotation of the ball was restricted around the vertical axis with a pin. The virtual reality environment was projected onto a toroidal screen covering approximately 240° horizontally and 100° vertically of the mouse's visual field, using a projector (Samsung SP-F10M) synchronized to the resonant scanner of the two-photon microscope such that visual stimulation occurs at the turnaround time of the resonant scanner, during which any photon reaching the PMT is discarded, thereby allowing for near-simultaneous stimulation-artifact-free imaging. The virtual environment consisted of an infinite corridor with walls patterned with vertical sinusoidal gratings with a spatial frequency of approximately 0.04 cycles per degree (*Leinweber et al., 2014*). In closed loop sessions, the locomotion of the mouse was coupled to movement along a virtual tunnel. In open loop sessions, we uncoupled the two and replayed the visual flow from a preceding closed loop session. In grating sessions, we presented full-field drifting gratings (0°, 45°, 90°, 270°, moving in either direction) in a pseudo-random sequence. Grating stimuli were presented for between 1 s and 8 s depending on the experiment. In the inter-stimulus interval, mice were shown a gray screen with average luminance matched to that of the grating stimuli. For optogenetic light stimulation, we used 1 s pulses (10 mW after the objective) of a red laser (637 nm) directed at visual cortex through a cranial window.

## Two-photon imaging

Two-photon calcium imaging was performed using custom-built microscopes (*Leinweber et al., 2014*). The illumination source was a tunable femtosecond laser (Insight, Spectra Physics or Chameleon, Coherent) tuned to 930 nm. Emission light was band-pass filtered using a 525/50 filter for GCaMP and a 607/70 filter for tdTomato/mCherry (Semrock) and detected using a GaAsP photomultiplier (H7422, Hamamatsu). Photomultiplier signals were amplified (DHPCA-100, Femto), digitized (NI5772, National Instruments) at 800 MHz, and band-pass filtered at 80 MHz using a digital Fourier-transform filter implemented in custom-written software on an FPGA (NI5772, National Instruments). The scanning system of the microscopes was based on a 12 kHz resonant scanner (Cambridge Technology). Images were acquired at a resolution of 750x400 pixels (60 Hz frame rate), and a piezo-electric linear actuator (P-726, Physik Instrumente) was used to move the objective (Nikon 16 x, 0.8 NA) in steps of 15 μm between frames to acquire images at 4 different depths. This resulted in an effective frame rate of 15 Hz. We imaged layer 2/3 at a depth of 100–250 μm, and layer 5 at a depth of 400–600 μm. The field of view was 375 μm × 300 μm.

## Widefield imaging

Widefield imaging experiments were conducted on a custom-build macroscope with commercially available objectives mounted face-to-face (Nikon 85 mm/f1.8 sample side, Nikon 50 mm/f1.4 sensor side). A 470 nm LED (Thorlabs) powered by a custom-build LED driver was used to excite GFP through an excitation filter (SP490, Thorlabs) reflected off a dichroic mirror (LP490, Thorlabs) placed parfocal to the objectives. The fluorescence was collected through a 525/50 emission filter on a sCMOS camera (PCO edge 4.2). LED illumination was adjusted with a collimator (Thorlabs SM2F32-A) to homogenously illuminate cortical surface through the cranial window. An Arduino board (Arduino Mega 2560) was used to synchronize LED onset with frame trigger signal of the camera. The duty cycle of the 470 nm LED was 90%. Images were acquired at 50 Hz or 100 Hz effective frame rate. Raw images were cropped on the sensor and data was stored to disk with custom-written LabVIEW (National Instruments) software, resulting in an effective pixel size of 60 μm$^2$ at a resolution of 1108 pixels × 1220 pixels (1.35 MP).

## Extraction of neuronal and GRAB activity

Two-photon calcium imaging data were processed as previously described (*Keller et al., 2012*) and all data analysis was done in MATLAB (MathWorks). Briefly, raw images were full-frame registered to correct for lateral brain motion. All data with visible z-motion were excluded. Neurons were manually selected based on mean and maximum fluorescence images. For GRAB imaging, neuropil and blood vessels were manually selected. Raw fluorescence traces were corrected for slow drift in fluorescence using an 8th-percentile filtering with a 66 s (or 1000 frames) window (*Dombeck et al., 2007*). $\Delta F/F_0$ traces were calculated as mean fluorescence in a selected region of every imaging frame, subtracted and normalized by the overall median fluorescence. No neuropil subtraction was done. All two-photon calcium and GRAB imaging data was acquired at 15 Hz.

For widefield macroscope imaging, raw movie data was manually registered across days by aligning subsequent mean projections of the data to the first recorded image sequence. ROIs were placed relative to readily identifiable anatomical landmarks that had been previously noted during cranial window surgery (see above). This resulted in the selection of six 20 pixels × 20 pixels ROIs per hemisphere. Of those ROIs, we calculated the activity as the $\Delta F/F_0$, wherein $F_0$ was the median fluorescence of the recording (approximately 30,000 frames in a 5 min recording). $\Delta F/F_0$ was corrected for slow fluorescence drift caused by thermal brightening of the LED using 8th percentile filtering with a 62.5 s moving window similar to what's described above for two-photon imaging.

## Extraction of blood vessel area

To measure the area of blood vessels, we manually identified rectangular ROIs encompassing the blood vessels and a surrounding margin of neuropil, as depicted in *Figure 4A and B*. We generated a trigger-averaged video from these ROIs by averaging the video across all instances of optogenetic stimulation under dark condition. We applied a threshold set at the 0.5th percentile of all pixel intensities in the video to binarize each frame. Subsequently, frame-wise we fitted an ellipse to the

connected pixels that fell below this threshold, isolating the blood vessel. The area of this ellipse served as an estimate of the blood vessel area.

## Data analysis

All data analysis was done using custom scripts written in MATLAB (MathWorks). To quantify the average population response traces, we first calculated the average event-triggered fluorescence trace for each ROI. The responses of all ROIs were then averaged and baseline-subtracted.

Locomotion onset was defined as running speed crossing a threshold of 0.25 cm/s for at least 1 s, while having been below the threshold during the preceding 1 s. The same criteria were used to define visual flow onsets in the open loop condition using visual flow speed. Visuomotor mismatch responses were probed by presenting brief 1 s full-field visual flow halts in the closed loop condition. For a visuomotor mismatch event to be included in analysis, mice had to be locomoting uninterrupted above threshold (0.25 cm/s from –0.5 s to +1 s after the event onset. Additionally, for a ROI to be included for analysis of the response to a particular event, it had to have at least three onsets to the event. The GFP response was baseline subtracted using a –0.5 s to 0 s window relative to onset. The same was –1 s to –0.5 s in case of locomotion onset to account for preparatory activity.

In *Figure 2C–E*, to compare GFP and GCaMP responses, we trial averaged the response of individual neurons and used the mean response over a time-window of 0 s to +2.0 s relative to locomotion or visuomotor mismatch onset, with a baseline subtraction window of –1 s to –0.5 s. For grating responses, we used the mean of the trial averaged response over +0.5 s to +3.0 s relative to stimulus onset, with a baseline subtraction window of –0.5 s to 0 s.

In *Figure 3*, for each neuron, we averaged the response from +0.5 s to +1.5 s relative to onset for individual locomotion onset and visuomotor mismatch trials and tested if the mean of this distribution is significantly different from 0 using a t-test with a chance threshold of 5%. In a similar analysis, the response window for grating responsive neurons was set at +0.5 s to +2.5 s to account for the 2 s long grating stimuli.

In *Figure 4*, the blood vessel diameter and the average GFP signal from both blood vessel ROIs and neuropil ROIs were smoothed with a 0.5 s moving average and the variance explained was computed on the –2.0 s to +6.0 s window relative to stimulus onset with a linear model (i.e. $R^2$, where R is the linear correlation coefficient between the two signals). In *Figure 4D*, data points were drawn as outliers if they were larger than 75th percentile +1.5*(75th percentile - 25th percentile) or smaller than 25th percentile - 1.5*(75th percentile - 25th percentile) of the distribution and were omitted from the figure.

In *Figure 7*, *Figure 7—figure supplement 1*, due to insufficient triggers, we relaxed the locomotion threshold for mismatch to 0.12 cm/s, and in *Figure 7—figure supplement 1*, we relaxed the sitting window before locomotion onset to –0.5 s to 0 s.

## Statistical analysis

All statistical information for the tests performed in the manuscript is provided in *Supplementary file 1*. Unless stated otherwise, the shading indicates the error in mean estimation to within 1 standard deviation. For analysis where the experimental unit was neurons (or ROIs), we used hierarchical bootstrap (*Saravanan et al., 2020*) for statistical testing due to the nested nature (neurons [or ROIs] and mice) of the data. Briefly, we trial averaged the data for the respective event per neuron (or ROI), and first resampled the data (with replacement) at the level of imaging sites, and then, from the selected sites, resampled for neurons. We then computed the mean of this bootstrap sample and repeated this N times to generate a bootstrap distribution of the mean estimate. For all statistical testing, the number of bootstrap samples (N) was 10,000, and for plotting bootstrap mean and SE response curves it was 1000. The bootstrap SE is the 68% confidence interval (1 SD, 15.8th percentile to 84.2nd percentile) in the bootstrap distribution of the mean.

## Acknowledgements

We thank all the members of the Keller lab for discussion and support. This project has received funding from the Swiss National Science Foundation (GBK), the Novartis Research Foundation (GBK), and the European Research Council (ERC) under the European Union's Horizon 2020 research and innovation programme (grant agreement No 865617) (GBK).

# Additional information

## Funding

| Funder | Grant reference number | Author |
|---|---|---|
| Swiss National Science Foundation | | Georg B Keller |
| European Research Council | 10.3030/865617 | Georg B Keller |
| Novartis Research Foundation | | Georg B Keller |

The funders had no role in study design, data collection and interpretation, or the decision to submit the work for publication.

## Author contributions

Baba Yogesh, Conceptualization, Resources, Data curation, Software, Formal analysis, Validation, Investigation, Visualization, Methodology, Writing – original draft, Project administration, Writing – review and editing; Matthias Heindorf, Resources, Software, Investigation, Methodology, Writing – review and editing; Rebecca Jordan, Resources, Investigation, Methodology, Writing – review and editing; Georg B Keller, Supervision, Funding acquisition, Writing – original draft, Writing – review and editing

## Author ORCIDs

Baba Yogesh ⓘ https://orcid.org/0000-0003-1224-6035
Georg B Keller ⓘ https://orcid.org/0000-0002-1401-0117

## Ethics

All animal procedures were approved by and carried out in accordance with the guidelines laid by the Veterinary Department of the Canton of Basel-Stadt, Switzerland. License 2573.

Reviewer #1 (Public review): https://doi.org/10.7554/eLife.104914.3.sa1
Reviewer #2 (Public review): https://doi.org/10.7554/eLife.104914.3.sa2
Reviewer #3 (Public review): https://doi.org/10.7554/eLife.104914.3.sa3
Author response https://doi.org/10.7554/eLife.104914.3.sa4

# Additional files

## Supplementary files

Supplementary file 1. Statistical information on all analysis.

Supplementary file 2. Number of mice in different experimental groups.

MDAR checklist

## Data availability

All data and code necessary to generate the figures of this paper are deposited here: https://doi.org/10.5281/zenodo.15198651.

The following dataset was generated:

| Author(s) | Year | Dataset title | Dataset URL | Database and Identifier |
|---|---|---|---|---|
| Yogesh B, Heindorf M, Jordan R, Keller GB | 2025 | Quantification of the effect of hemodynamic occlusion in two-photon imaging | https://doi.org/10.5281/zenodo.15198651 | Zenodo, 10.5281/zenodo.15198651 |

The following previously published dataset was used:

| Author(s) | Year | Dataset title | Dataset URL | Database and Identifier |
|---|---|---|---|---|
| Yogesh B, Keller GB | 2024 | Cholinergic input to mouse visual cortex signals a movement state and acutely enhances layer 5 responsiveness | https://doi.org/10.5281/zenodo.12632129 | Zenodo, 10.5281/zenodo.12632129 |

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
