## [Editor Report · eLife Assessment]

This **important** study conducted experiments to quantify how neural activity independent changes in fluorescence might affect two-photon recordings when using diverse sensors. The researchers found a widespread presence of neural-activity-independent artifacts in two-photon imaging and provide **convincing** evidence that these artifacts are most likely caused by hemodynamic occlusion. Their findings underscore the importance of accounting for these artifacts when interpreting functional two-photon recordings.

---

## [Referee Report · Reviewer #1 (Public review)]

Summary:

Fluorescence imaging has become an increasingly popular technique for monitoring neuronal activity and neurotransmitter concentrations in the living brain. However, factors such as brain motion and changes in blood flow and oxygenation can introduce significant artifacts, particularly when activity-dependent signals are small. Yogesh et al. quantified these effects using GFP, an activity-independent marker, under two-photon and wide-field imaging conditions in awake behaving mice. They report significant GFP responses across various brain regions, layers, and behavioral contexts, with magnitudes comparable to those of commonly used activity sensors. These data highlight the need for robust control strategies and careful interpretation of fluorescence functional imaging data.

Strengths:

The effect of hemodynamic occlusion in two-photon imaging has been previously demonstrated in sparsely labeled neurons in V1 of anesthetized animals (see Shen and Kara et al., Nature Methods, 2012). The present study builds on these findings by imaging a substantially larger population of neurons in awake, behaving mice across multiple cortical regions, layers, and stimulus conditions. The experiments are extensive, the statistical analyses are rigorous, and the results convincingly demonstrate significant GFP responses that must be accounted for in functional imaging experiments.

In the revised version, the authors have provided further methodological details that were lacking in the previous version, expanded discussions regarding alternative explanations of these GFP responses as well as potential mitigation strategies. They also added a quantification of brain motion (Fig. S5) and the fraction of responsive neurons when conducting the same experiment using GCaMP6f (Fig. 3D-3F), among other additional information.

Weaknesses:

(1) The authors have now included a detailed methodology for blood vessel area quantification, where they detect blood vessels as dark holes in GFP images and measure vessel area by counting pixels below a given intensity threshold (line 437-443). However, this approach has a critical caveat: any unspecific decrease in image fluorescence will increase the number of pixels below the threshold, leading to an apparent increase in blood vessel area, even when the actual vessel size remains unchanged. As a result, this method inherently introduces a positive correlation between fluorescence decrease and vessel dilation, regardless of whether such a relationship truly exists.

To address this issue, I recommend labelling blood vessels with an independent marker, such as a red fluorescence dye injected into the bloodstream. This approach would allow vessel dilation to be assessed independently of GFP fluorescence -- dilation would cause opposite fluorescence changes in the green and red channels (i.e., a decrease in green due to hemodynamic occlusion and an increase in red due to the expanding vessel area). In my opinion, only when such ani-correlation is observed can one reliably infer a relationship between GFP signal changes and blood vessel dynamics.

Because this relationship is central to the author's conclusion regarding the nature of the observed GFP signals, including this experiment would greatly strengthen the paper's conclusion.

(2) Regarding mitigation strategy, the authors advocate repeating key functional imaging experiments using GFP, and state that their aim here is to provide a control for their 2012 study (Keller et al., Neuron). Given this goal, I find it important to discuss how these new findings impact the interpretation of their 2012 results, particularly given the large GFP responses observed.

For example, Keller et al. (2012) concluded that visuomotor mismatch strongly drives V1 activity (Fig. 3A in that study). However, in the present study, mismatch fails to produce any hemodynamic/GFP response (Fig. 3A, 3B, rightmost bar), and the corresponding calcium response is also the weakest among the three tested conditions (Fig. 3D). How do these findings affect their 2012 conclusions?

Similarly, the present study shows that GFP reveals twice as many responsive neurons as GCaMP during locomotion (Fig. 3A vs. Fig. 3D, "running"). Does this mean that their 2012 conclusions regarding locomotion-induced calcium activity need reconsideration? Given that more neurons responded with GFP than with GCaMP, the authors should clarify whether they still consider GCaMP a reliable tool for measuring brain activity during locomotion.

(3) More generally, the author should discuss how functional imaging data should be interpreted going forward, given the large GFP responses reported here. Even when key experiments are repeated using GFP, it is not entirely clear how one could reliably estimate underlying neuronal activity from the observed GFP and GCaMP responses.

For example, consider the results in Fig. 3A vs. 3D: how should one assess the relative strength of neuronal activity elicited by running, grating, or visuomotor mismatch? Does mismatch produce the strongest neuronal activity, since it is least affected by the hemodynamic/GFP confounds (Fig. 3A)? Or does mismatch actually produce the weakest neuronal activity, given that both its hemodynamic and calcium responses are the smallest?

In my opinion, such uncertainty makes it difficult to robustly interpret functional imaging results. Simply repeating experiments with GFP does not fully resolve this issue, as it does not provide a clear framework for quantifying the underlying neuronal activity. Does this suggest a need for a better mitigation strategy? What could these strategies be?

In my opinion, addressing these questions is critical not only for the authors' own work but also for the broader field to ensure a robust and reliable interpretation of functional imaging data.

(4) The authors now discuss various alternative sources of the observed GFP signals. However, I feel that they often appear to dismiss these possibilities too quickly, rather than appreciating their true potential impacts (see below).

For example, the authors argue that brain movement cannot explain their data, as movement should only result in a decrease in observed fluorescence. However, while this might hold for x-y motion, movement in the axial (z) direction can easily lead to both fluorescence increase and decrease. Neurons are not always precisely located at the focal plane -- some are slightly above or below. Axial movement in a given direction will bring some cells into focus while moving others out of focus, leading to fluorescence changes in both directions, exactly as observed in the data (see Fig. S2).

Furthermore, the authors state that they discard data with 'visible' z-motion. However, subtle axial movements that escape visual detection could still cause fluorescence fluctuations on the order of a few percent, comparable to the reported signal amplitudes.

Finally, the authors state that "brain movement kinematics are different in shape than the GFP responses we observe". However, this appears to contradict what they show in Fig. 2A. Specifically, the first example neuron exhibits fast GFP transients locked to running onset, with rapid kinematics closely matching the movement speed signals in Fig. S5A. These fast transients are incompatible with slower blood vessel area signals (Fig. 4), suggesting that alternative sources could contribute significantly.

In sum, the possibility that alternative signal sources could significantly contribute should be taken seriously and more thoroughly discussed.

(5) The authors added a quantification of brain movement (Fig. S5) and claim that they "only find detectable brain motion during locomotion onsets and not the other stimuli." However, Fig. S5 presents brain 'velocity' rather than 'displacement'. A constant (non-zero) velocity in Fig. S5 B-D indicates that the brain continues to move over time, potentially leading to significant displacement from its initial position across all conditions. While displacement in the x-y plane are corrected, similar displacement in the z direction likely occurs concurrently and cannot be easily accounted for. To assess this possibility, the authors should present absolute displacement relative to pre-stimulus frames, as displacement -- not velocity -- determines the size of movement-related fluorescence changes.

(6) In line 132-133, the authors draw an analogy between the effect of hemodynamic occlusion and liquid crystal display (LCD) function. However, there are fundamental differences between the two. LCDs modulate light transmission by rotating the polarization of light, which then passes through a crossed polarizer. In contrast, hemodynamic occlusion alters light transmission by changing the number and absorbance properties of hemoglobin. Additionally, LCDs do not involve 'emission' light - back-illumination travels through the liquid crystal layer only once, whereas hemodynamic occlusion affects both incoming excitation light and the emitted fluorescence. Given these fundamental differences, the LCD analogy may not be entirely appropriate.

---

## [Referee Report · Reviewer #2 (Public review)]

- Approach

In this study, Yogesh et al. aimed at characterizing hemodynamic occlusion in two photon imaging, where its effects on signal fluctuations are underappreciated compared to that in wide field imaging and fiber photometry. The authors used activity-independent GFP fluorescence, GCaMP and GRAB sensors for various neuromodulators in two-photon and widefield imaging during a visuomotor context to evaluate the extent of hemodynamic occlusion in V1 and ACC. They found that the GFP responses were comparable in amplitude to smaller GCaMP responses, though exhibiting context-, cortical region-, and depth-specific effects. After quantifying blood vessel diameter change and surrounding GFP responses, they argued that GFP responses were highly correlated with changes in local blood vessel size. Furthermore, when imaging with GRAB sensors for different neuromodulators, they found that sensors with lower dynamic ranges such as GRAB-DA1m, GRAB-5HT1.0, and GRAB-NE1m exhibited responses most likely masked by the hemodynamic occlusion, while a sensor with larger SNR, GRAB-ACh3.0, showed much more distinguishable responses from blood vessel change. They thoroughly investigate other factors that could contribute to these signals and demonstrate hemodynamic occlusion is the primary cause.

- Impact of revision

This is an important update to the initial submission, adding much supplemental imaging and population data that provide greater detail to the analyses and increase the confidence in the authors conclusions.

Specifically, inclusion of the supplemental figures 1 and 2 showing GFP expression across multiple regions and the fluorescence changes of thousands of individual neurons provides a clearer picture of how these effects are distributed across the population. Characterization of brain motion across stimulation conditions in supplemental figure 5 provides strong evidence that the fluorescence changes observed in many of the conditions are unlikely to be primarily due to brain motion associated imaging artifacts. The role of vascular area on fluorescence is further supported by addition of new analyses on vasoconstriction leading to increased fluorescence in Figures 4C1-4, complementing the prior analyses of vasodilation.

The expansion of the discussion on other factors that could lead to these changes is thorough and welcome. The arguments against pH playing a factor in fluorescence changes of GFP, due to insensitivity to changes in the expected pH range are reasonable, as are the other discussed potential factors.

With respect to the author's responses to prior critique, we agree that activity dependent hemodynamic occlusion is best investigated under awake conditions. Measurement of these dynamics under anesthesia could lead to an underestimation of their effects. Isoflurane anesthesia causes significant vasodilation and a large reduction in fluorescence intensity in non-functional mutant GRABs. This could saturate or occlude activity dependent effects.

- Strengths

This work is of broad interest to two photon imaging users and GRAB developers and users. It thoroughly quantifies the hemodynamic driven GFP response and compares it to previously published GCaMP data in a similar context, and illustrates the contribution of hemodynamic occlusion to GFP and GRAB responses by characterizing the local blood vessel diameter and fluorescence change. These findings provide important considerations for the imaging community and a sobering look at the utility of these sensors for cortical imaging.

Importantly, they draw clear distinctions between the temporal dynamics and amplitude of hemodynamic artifacts across cortical regions and layers. Moreover, they show context dependent (Dark versus during visual stimuli) effects on locomotion and optogenetic light-triggered hemodynamic signals.

The authors suggest that signal to noise ratio of an indicator likely affects the ability to separate hemodynamic response from the underlying fluorescence signal. With a new analysis (Supplemental Figure 4) They show that the relative degree of background fluorescence does not affect the size of the artifact.

Most of the first generation neuromodulator GRAB sensors showed relatively small responses, comparable to blood vessel changes in two photon imaging, which emphasizes a need for improved the dynamic range and response magnitude for future sensors and encourages the sensor users to consider removing hemodynamic artifacts when analyzing GRAB imaging data.

- Weaknesses

The largest weakness of the paper remains that, while they convincingly quantify hemodynamic artifacts across a range of conditions, they provide limited means of correcting for them. However they now discuss the relative utility of some hemodynamic correction methods (e.g. from Ocana-Santero et al., 2024).

The paper attributes the source of 'hemodynamic occlusion' primarily to blood vessel dilation, but leaves unanswered how much may be due to shifts in blood oxygenation. Figure 4 directly addresses the question of how much of the signal can be attributed to occlusion by measuring the blood vessel dilation, and has been improved by now showing positive fluorescence effects with vasoconstriction. They now also discuss the potential impact of oxygenation.

Along these lines, the authors carefully quantified the correlation between local blood vessel diameter and GFP response (or neuropil fluorescence vs blood vessel fluorescence with GRAB sensors). We are left to wonder to what extent does this effect depend on proximity to the vessels? Do GFP/ GRAB responses decorrelate from blood vessel activity in neurons further from vessels (refer to Figure 5A and B in Neyhart et al., Cell Reports 2024)? The authors argue that the primary impact of occlusion is from blood vessels above the plane of imaging, but without a vascular reconstruction, their evidence for this is anecdotal.

The choice of ACC as the frontal region provides a substantial contrast in location, brain movement, and vascular architecture as compared to V1. As the authors note, ACC is close to the superior sagittal sinus and thus is the region where the largest vascular effects are likely to occur. A less medial portion of M2 may have been a more appropriate comparison. The authors now include example imaging fields for ACC and interesting out-of-plane vascular examples in the supplementary figures that help assess these impacts.

-Overall Assessment

This paper is an important contribution to our understanding of how hemodynamic artifacts may corrupt GRAB and calcium imaging, even in two-photon imaging modes. While it would be wonderful if the authors were able to demonstrate a reliable way to correct for hemodynamic occlusion which did not rely on doing the experiments over with a non-functional sensor or fluorescent protein, the careful measurement and reporting of the effects here is, by itself, a substantial contribution to the field of neural activity imaging. It's results are of importance to anyone conducting two-photon or widefield imaging with calcium and GRAB sensors and deserves the attention of the broader neuroscience and in-vivo imaging community.

---

## [Referee Report · Reviewer #3 (Public review)]

Summary:

In this study, the authors aimed to investigate if hemodynamic occlusion contributes to fluorescent signals measured with two-photon microscopy. For this, they image the activity-independent fluorophore GFP in 2 different cortical areas, at different cortical depths and in different behavioral conditions. They compare the evoked fluorescent signals with those obtained with calcium sensors and neuromodulator sensors and evaluate their relationship to vessel diameter as a readout of blood flow.

They find that GFP fluorescence transients are comparable to GCaMP6f stimuli-evoked signals in amplitude, although they are generally smaller. Yet, they are significant even at the single neuronal level. They show that GFP fluorescence transients resemble those measured with the dopamine sensor GRAB-DA1m and the serotonin sensor GRAB-5HT1.0 in amplitude an nature, suggesting that signals with these sensors are dominated by hemodynamic occlusion.Moreover, the authors perform similar experiments with wide-field microscopy which reveals the similarity between the two methods in generating the hemodynamic signals. Together the evidence presented calls for the development and use of high dynamic range sensors to avoid measuring signals that have another origin from the one intended to measure. In the meantime, the evidence highlights the need to control for those artifacts such as with the parallel use of activity independent fluorophores.

Strengths:

- Comprehensive study comparing different cortical regions in diverse behavioral settings in controlled conditions.

- Comparison to the state-of-the-art, i.e. what has been demonstrated with wide-field microscopy.

- Comparison to diverse activity-dependent sensors, including the widely used GCaMP.

Comments on revisions:

The authors have addressed my concerns well. I have no further comments.

---

## [Author Response]

The following is the authors’ response to the current reviews.

We thank you for the time you took to review our work and for your feedback! We have made only minor changes in this submission and primarily wanted to respond to the concerns raised by reviewer 1.

**Reviewer #1 (Public review):**
Summary:Fluorescence imaging has become an increasingly popular technique for monitoring neuronal activity and neurotransmitter concentrations in the living brain. However, factors such as brain motion and changes in blood flow and oxygenation can introduce significant artifacts, particularly when activitydependent signals are small. Yogesh et al. quantified these effects using GFP, an activity-independent marker, under two-photon and wide-field imaging conditions in awake behaving mice. They report significant GFP responses across various brain regions, layers, and behavioral contexts, with magnitudes comparable to those of commonly used activity sensors. These data highlight the need for robust control strategies and careful interpretation of fluorescence functional imaging data.Strengths:The effect of hemodynamic occlusion in two-photon imaging has been previously demonstrated in sparsely labeled neurons in V1 of anesthetized animals (see Shen and Kara et al., Nature Methods, 2012). The present study builds on these findings by imaging a substantially larger population of neurons in awake, behaving mice across multiple cortical regions, layers, and stimulus conditions. The experiments are extensive, the statistical analyses are rigorous, and the results convincingly demonstrate significant GFP responses that must be accounted for in functional imaging experiments.In the revised version, the authors have provided further methodological details that were lacking in the previous version, expanded discussions regarding alternative explanations of these GFP responses as well as potential mitigation strategies. They also added a quantification of brain motion (Fig. S5) and the fraction of responsive neurons when conducting the same experiment using GCaMP6f (Fig. 3D-3F), among other additional information.Weaknesses:(1) The authors have now included a detailed methodology for blood vessel area quantification, where they detect blood vessels as dark holes in GFP images and measure vessel area by counting pixels below a given intensity threshold (line 437-443). However, this approach has a critical caveat: any unspecific decrease in image fluorescence will increase the number of pixels below the threshold, leading to an apparent increase in blood vessel area, even when the actual vessel size remains unchanged. As a result, this method inherently introduces a positive correlation between fluorescence decrease and vessel dilation, regardless of whether such a relationship truly exists.To address this issue, I recommend labelling blood vessels with an independent marker, such as a red fluorescence dye injected into the bloodstream. This approach would allow vessel dilation to be assessed independently of GFP fluorescence -- dilation would cause opposite fluorescence changes in the green and red channels (i.e., a decrease in green due to hemodynamic occlusion and an increase in red due to the expanding vessel area). In my opinion, only when such ani-correlation is observed can one reliably infer a relationship between GFP signal changes and blood vessel dynamics.Because this relationship is central to the author's conclusion regarding the nature of the observed GFP signals, including this experiment would greatly strengthen the paper's conclusion.

This is correct – a more convincing demonstration that blood vessels dilate or constrict anticorrelated with apparent GFP fluorescence would be a separate blood vessel marker. However, we don’t think this experiment is worth doing, as it is also not conclusive in the sense the reviewer may have in mind. The anticorrelation does not mean that occlusion drives all of the observed effect. Our main argument is instead that there is no other potential source than hemodynamic occlusion with sufficient strength that we can think of. The experiment one would want to do is block hemodynamic changes and demonstrate that the occlusion explains all of the observed changes.

(2) Regarding mitigation strategy, the authors advocate repeating key functional imaging experiments using GFP, and state that their aim here is to provide a control for their 2012 study (Keller et al., Neuron). Given this goal, I find it important to discuss how these new findings impact the interpretation of their 2012 results, particularly given the large GFP responses observed.

We are happy to discuss how the conclusions of our own work are influenced by this (see more details below), but the important response of the field should probably be to revisit the conclusions of a variety of papers published in the last two decades. This goes far beyond what we can do here.

For example, Keller et al. (2012) concluded that visuomotor mismatch strongly drives V1 activity (Fig. 3A in that study). However, in the present study, mismatch fails to produce any hemodynamic/GFP response (Fig. 3A, 3B, rightmost bar), and the corresponding calcium response is also the weakest among the three tested conditions (Fig. 3D). How do these findings affect their 2012 conclusions?

The average calcium response of L2/3 neurons to visuomotor mismatch is probably roughly similar to the average calcium response at locomotion onset (both are on the order of 1% to 5%, depending on indicator, dataset, etc.). In the Keller et al. (2012) paper, locomotion onset was about 1.5% and mismatch about 3% (see Figure 3A in that paper). What we quantify in Figure 3 of the paper here is the fraction of responsive neurons. Thus, mismatch drives strong responses in a small subset of neurons (approx. 10%), while locomotion drives a combination of a weak responses in a large fraction of the neurons (roughly 70%) and also large responses in a subset of neurons. A strong signal in a subset of neurons is what one would expect from a neuronal response, a weak signal from many neurons would be indicative of a contaminating signal. This all appears consistent.

Regarding influencing the conclusions of earlier work, the movement related signals described in the Keller et al. (2012) paper are probably overestimated, but are also apparent in electrophysiological recordings (Saleem et al., 2013). Thus, the locomotion responses reported in the Keller et al. (2012) paper are likely too high, but locomotion related responses in V1 are very likely real. The only conclusion we draw in the Keller et al. 2012 paper on the strength of the locomotion related responses is that they are smaller than mismatch responses (this conclusion is unaffected by hemodynamic contamination). In addition, the primary findings of the Keller et al. (2012) paper are all related to mismatch, and these conclusions are unaffected.

Similarly, the present study shows that GFP reveals twice as many responsive neurons as GCaMP during locomotion (Fig. 3A vs. Fig. 3D, "running"). Does this mean that their 2012 conclusions regarding locomotion-induced calcium activity need reconsideration? Given that more neurons responded with GFP than with GCaMP, the authors should clarify whether they still consider GCaMP a reliable tool for measuring brain activity during locomotion.

Comparisons of the fraction of significantly responsive neurons between GFP and GCaMP are not straightforward to interpret. One needs to factor in the difference in signal to noise between the two sensors. (Please note, we added the GCaMP responses here upon request of the reviewers). Note, there is nothing inherently wrong with the data, and comparisons within dataset are easily made (e.g. more grating responsive neurons than running responsive neurons in GCaMP, and vice versa with GFP). The comparison across datasets is not as straightforward as we define “responsive neurons” using a statistical test that compares response to baseline activity for each neuron. GFP labelled neurons are very bright and occlusion can easily be detected. Baseline fluorescence in GCaMP recordings is much lower and often close to or below the noise floor of the data (i.e. we only see the cells when they are active). Thus occlusion in GCaMP recordings is preferentially visible for cells that have high baseline fluorescence. Thus, in the GCaMP data we are likely underestimating the fraction of responsive neurons.

Regarding whether GCaMP (or any other fluorescence indicator used in vivo) is a reliable tool, we are not sure we understand. Whenever possible, fluorescence-sensor based measurements should be corrected for hemodynamic contamination – to quantify locomotion related signals this will be more difficult than e.g. for mismatch, but that does not mean it is not reliable.

(3) More generally, the author should discuss how functional imaging data should be interpreted going forward, given the large GFP responses reported here. Even when key experiments are repeated using GFP, it is not entirely clear how one could reliably estimate underlying neuronal activity from the observed GFP and GCaMP responses.

We are not sure we have a good answer to this question. The strategy for addressing this problem will depend on the specifics of the experiment, and the claims. Take the case of mismatch. Here we have strong calcium responses and no evidence of GFP responses. We would argue that this is reasonable evidence that the majority of the mismatch driven GCaMP signal is likely neuronal. For locomotion onsets, both GFP and GCaMP signals go in the same direction on average. Then one could use a response amplitude distribution comparison to conservatively exclude all neurons with a GCaMP amplitude lower than e.g. the 99th percentile of the GFP response. Etc. But we don’t think there is an easy generalizable fix for this problem.

For example, consider the results in Fig. 3A vs. 3D: how should one assess the relative strength of neuronal activity elicited by running, grating, or visuomotor mismatch? Does mismatch produce the strongest neuronal activity, since it is least affected by the hemodynamic/GFP confounds (Fig. 3A)? Or does mismatch actually produce the weakest neuronal activity, given that both its hemodynamic and calcium responses are the smallest?

See above, the reviewer may be confounding “response strength” with “fraction of responsive neurons” here. Regarding the relationship between neuronal activity and hemodynamics, it is very likely not just the average activity of all neurons, but a specific subset that drives blood vessel constriction and dilation. This would of course be a very interesting question to answer for the interpretation of hemodynamic based measurements of brain activity, like fMRI, but goes beyond the aim of the current paper.

In my opinion, such uncertainty makes it difficult to robustly interpret functional imaging results. Simply repeating experiments with GFP does not fully resolve this issue, as it does not provide a clear framework for quantifying the underlying neuronal activity. Does this suggest a need for a better mitigation strategy? What could these strategies be?

If the reviewer has a good idea - we would be all ears. We don’t have a better idea currently.

In my opinion, addressing these questions is critical not only for the authors' own work but also for the broader field to ensure a robust and reliable interpretation of functional imaging data.

We agree, having a solution to this problem would be important – we just don’t have one.

(4) The authors now discuss various alternative sources of the observed GFP signals. However, I feel that they often appear to dismiss these possibilities too quickly, rather than appreciating their true potential impacts (see below).For example, the authors argue that brain movement cannot explain their data, as movement should only result in a decrease in observed fluorescence. However, while this might hold for x-y motion, movement in the axial (z) direction can easily lead to both fluorescence increase and decrease. Neurons are not always precisely located at the focal plane -- some are slightly above or below. Axial movement in a given direction will bring some cells into focus while moving others out of focus, leading to fluorescence changes in both directions, exactly as observed in the data (see Fig. S2).

The reviewer is correct that z-motion can result in an increase of apparent fluorescence (just like x-y motion can as well). On average however, just like with x-y motion, z-motion will always result in a decrease. This assumes that the user selecting regions of interest (the outlines of cells used to quantify fluorescence), will select these such that the distribution of cells selected centers on the zplane of the image. Thus, the distribution of z-location of the cell relative to the imaging plane will be some Gaussian like distribution centered on the z-plane of the image (with half the cell above the zplane and half below). Because the peak of the distribution is located on the z-plane at rest, any zmovement, up or down, will move away from the peak of the distribution (i.e. most cells will decrease in fluorescence). This is the same argument as for why x-y motion always results in decreases (assuming the user selects regions of interest centered on the location of the cells at rest).

Furthermore, the authors state that they discard data with 'visible' z-motion. However, subtle axial movements that escape visual detection could still cause fluorescence fluctuations on the order of a few percent, comparable to the reported signal amplitudes.

Correct, but as explained above, z-motion will always result in average decreases of average fluorescence as explained above.

Finally, the authors state that "brain movement kinematics are different in shape than the GFP responses we observe". However, this appears to contradict what they show in Fig. 2A. Specifically, the first example neuron exhibits fast GFP transients locked to running onset, with rapid kinematics closely matching the movement speed signals in Fig. S5A. These fast transients are incompatible with slower blood vessel area signals (Fig. 4), suggesting that alternative sources could contribute significantly.

We meant population average responses here. We have clarified this. Some of the signals we observed do indeed look like they could be driven by movement artifacts (whole brain motion, or probably more likely blood vessel dilation driven tissue distortion). We show this neuron to illustrate that this can also happen. However, to illustrate that this is a rare event we also show the entire distribution of peak amplitudes and the position in the distribution this neuron is from.

In sum, the possibility that alternative signal sources could significantly contribute should be taken seriously and more thoroughly discussed.

All possible sources (we could think of) are explicitly discussed (in roughly equal proportion). Nevertheless, the reviewer is correct that our focus here is almost exclusively on the what we think is the primary source of the problem. Given that – in my experience – this is also the one least frequently considered, I think the emphasis on – what we think is – the primary contributor is warranted.

(5) The authors added a quantification of brain movement (Fig. S5) and claim that they "only find detectable brain motion during locomotion onsets and not the other stimuli." However, Fig. S5 presents brain 'velocity' rather than 'displacement'. A constant (non-zero) velocity in Fig. S5 B-D indicates that the brain continues to move over time, potentially leading to significant displacement from its initial position across all conditions. While displacement in the x-y plane are corrected, similar displacement in the z direction likely occurs concurrently and cannot be easily accounted for. To assess this possibility, the authors should present absolute displacement relative to pre-stimulus frames, as displacement -- not velocity -- determines the size of movement-related fluorescence changes.

We use brain velocity here as a natural measure when using frame times as time bins. The problem with using a signed displacement is that if different running onsets move the brain in opposing directions, this can average out to zero. To counteract this, one can take the absolute displacement in a response window away from the position in a baseline time window. If this is done with time bins that correspond to frame times, this just becomes displacement per frame, i.e. velocity. Using absolute changes in displacement (i.e. velocity) is more sensitive than signed displacement. The responses for signed displacement are shown below (Author response image 1), but given that we are averaging signed quantities here, the average is not interpretable.

**Author response image 1. sa4fig1:** Average signed brain displacement.

Regarding a constant drift, the reviewer might be misled by the fact that the baseline brain velocity is roughly 1 pixel per frame. The registration algorithm works in integer number of pixels only. 1 pixel per frame corresponds roughly to the noise floor of the registration algorithm. Registrations are done independently for each frame. As a consequence, the registration oscillates between a shift of 17 and 18 pixels – frame by frame – if the actual shift is somewhere between 17 and 18 pixels. This “jitter” results in a baseline brain velocity of about 1 pixel per frame.

(6) In line 132-133, the authors draw an analogy between the effect of hemodynamic occlusion and liquid crystal display (LCD) function. However, there are fundamental differences between the two. LCDs modulate light transmission by rotating the polarization of light, which then passes through a crossed polarizer. In contrast, hemodynamic occlusion alters light transmission by changing the number and absorbance properties of hemoglobin. Additionally, LCDs do not involve 'emission' light - backillumination travels through the liquid crystal layer only once, whereas hemodynamic occlusion affects both incoming excitation light and the emitted fluorescence. Given these fundamental differences, the LCD analogy may not be entirely appropriate.

The mechanism of occlusion is, as the reviewer correctly points out, different for an LCD. In both cases however, there is a variable occluder between a light source and an observer. The fact that with hemodynamic occlusion the light passes through the occluder twice (excitation and emission) does not appear to hamper the analogy to us. We have rephrased to highlight the time varying occlusion part.

**Reviewer #2 (Public review):**
- ApproachIn this study, Yogesh et al. aimed at characterizing hemodynamic occlusion in two photon imaging, where its effects on signal fluctuations are underappreciated compared to that in wide field imaging and fiber photometry. The authors used activity-independent GFP fluorescence, GCaMP and GRAB sensors for various neuromodulators in two-photon and widefield imaging during a visuomotor context to evaluate the extent of hemodynamic occlusion in V1 and ACC. They found that the GFP responses were comparable in amplitude to smaller GCaMP responses, though exhibiting context-, cortical region-, and depth-specific effects. After quantifying blood vessel diameter change and surrounding GFP responses, they argued that GFP responses were highly correlated with changes in local blood vessel size. Furthermore, when imaging with GRAB sensors for different neuromodulators, they found that sensors with lower dynamic ranges such as GRAB-DA1m, GRAB-5HT1.0, and GRAB-NE1m exhibited responses most likely masked by the hemodynamic occlusion, while a sensor with larger SNR, GRAB-ACh3.0, showed much more distinguishable responses from blood vessel change. They thoroughly investigate other factors that could contribute to these signals and demonstrate hemodynamic occlusion is the primary cause.- Impact of revisionThis is an important update to the initial submission, adding much supplemental imaging and population data that provide greater detail to the analyses and increase the confidence in the authors conclusions.Specifically, inclusion of the supplemental figures 1 and 2 showing GFP expression across multiple regions and the fluorescence changes of thousands of individual neurons provides a clearer picture of how these effects are distributed across the population. Characterization of brain motion across stimulation conditions in supplemental figure 5 provides strong evidence that the fluorescence changes observed in many of the conditions are unlikely to be primarily due to brain motion associated imaging artifacts. The role of vascular area on fluorescence is further supported by addition of new analyses on vasoconstriction leading to increased fluorescence in Figures 4C1-4, complementing the prior analyses of vasodilation.The expansion of the discussion on other factors that could lead to these changes is thorough and welcome. The arguments against pH playing a factor in fluorescence changes of GFP, due to insensitivity to changes in the expected pH range are reasonable, as are the other discussed potential factors.With respect to the author's responses to prior critique, we agree that activity dependent hemodynamic occlusion is best investigated under awake conditions. Measurement of these dynamics under anesthesia could lead to an underestimation of their effects. Isoflurane anesthesia causes significant vasodilation and a large reduction in fluorescence intensity in non-functional mutant GRABs. This could saturate or occlude activity dependent effects.- StrengthsThis work is of broad interest to two photon imaging users and GRAB developers and users. It thoroughly quantifies the hemodynamic driven GFP response and compares it to previously published GCaMP data in a similar context, and illustrates the contribution of hemodynamic occlusion to GFP and GRAB responses by characterizing the local blood vessel diameter and fluorescence change. These findings provide important considerations for the imaging community and a sobering look at the utility of these sensors for cortical imaging.Importantly, they draw clear distinctions between the temporal dynamics and amplitude of hemodynamic artifacts across cortical regions and layers. Moreover, they show context dependent (Dark versus during visual stimuli) effects on locomotion and optogenetic light-triggered hemodynamic signals.The authors suggest that signal to noise ratio of an indicator likely affects the ability to separate hemodynamic response from the underlying fluorescence signal. With a new analysis (Supplemental Figure 4) They show that the relative degree of background fluorescence does not affect the size of the artifact.Most of the first generation neuromodulator GRAB sensors showed relatively small responses, comparable to blood vessel changes in two photon imaging, which emphasizes a need for improved the dynamic range and response magnitude for future sensors and encourages the sensor users to consider removing hemodynamic artifacts when analyzing GRAB imaging data.- WeaknessesThe largest weakness of the paper remains that, while they convincingly quantify hemodynamic artifacts across a range of conditions, they provide limited means of correcting for them. However they now discuss the relative utility of some hemodynamic correction methods (e.g. from Ocana-Santero et al., 2024).The paper attributes the source of 'hemodynamic occlusion' primarily to blood vessel dilation, but leaves unanswered how much may be due to shifts in blood oxygenation. Figure 4 directly addresses the question of how much of the signal can be attributed to occlusion by measuring the blood vessel dilation, and has been improved by now showing positive fluorescence effects with vasoconstriction. They now also discuss the potential impact of oxygenation.Along these lines, the authors carefully quantified the correlation between local blood vessel diameter and GFP response (or neuropil fluorescence vs blood vessel fluorescence with GRAB sensors). We are left to wonder to what extent does this effect depend on proximity to the vessels? Do GFP/ GRAB responses decorrelate from blood vessel activity in neurons further from vessels (refer to Figure 5A and B in Neyhart et al., Cell Reports 2024)? The authors argue that the primary impact of occlusion is from blood vessels above the plane of imaging, but without a vascular reconstruction, their evidence for this is anecdotal.The choice of ACC as the frontal region provides a substantial contrast in location, brain movement, and vascular architecture as compared to V1. As the authors note, ACC is close to the superior sagittal sinus and thus is the region where the largest vascular effects are likely to occur. A less medial portion of M2 may have been a more appropriate comparison. The authors now include example imaging fields for ACC and interesting out-of-plane vascular examples in the supplementary figures that help assess these impacts.-Overall AssessmentThis paper is an important contribution to our understanding of how hemodynamic artifacts may corrupt GRAB and calcium imaging, even in two-photon imaging modes. While it would be wonderful if the authors were able to demonstrate a reliable way to correct for hemodynamic occlusion which did not rely on doing the experiments over with a non-functional sensor or fluorescent protein, the careful measurement and reporting of the effects here is, by itself, a substantial contribution to the field of neural activity imaging. It's results are of importance to anyone conducting two-photon or widefield imaging with calcium and GRAB sensors and deserves the attention of the broader neuroscience and invivo imaging community.

We agree with this assessment.

**Reviewer #3 (Public review)**:Summary:In this study, the authors aimed to investigate if hemodynamic occlusion contributes to fluorescent signals measured with two-photon microscopy. For this, they image the activity-independent fluorophore GFP in 2 different cortical areas, at different cortical depths and in different behavioral conditions. They compare the evoked fluorescent signals with those obtained with calcium sensors and neuromodulator sensors and evaluate their relationship to vessel diameter as a readout of blood flow.They find that GFP fluorescence transients are comparable to GCaMP6f stimuli-evoked signals in amplitude, although they are generally smaller. Yet, they are significant even at the single neuronal level. They show that GFP fluorescence transients resemble those measured with the dopamine sensor GRABDA1m and the serotonin sensor GRAB-5HT1.0 in amplitude an nature, suggesting that signals with these sensors are dominated by hemodynamic occlusion. Moreover, the authors perform similar experiments with wide-field microscopy which reveals the similarity between the two methods in generating the hemodynamic signals. Together the evidence presented calls for the development and use of high dynamic range sensors to avoid measuring signals that have another origin from the one intended to measure. In the meantime, the evidence highlights the need to control for those artifacts such as with the parallel use of activity independent fluorophores.Strengths:- Comprehensive study comparing different cortical regions in diverse behavioral settings in controlled conditions.- Comparison to the state-of-the-art, i.e. what has been demonstrated with wide-field microscopy.- Comparison to diverse activity-dependent sensors, including the widely used GCaMP.Comments on revisions:The authors have addressed my concerns well. I have no further comments.

We agree with this assessment.

The following is the authors’ response to the original reviews

The major changes to the manuscript are:

(1) Re-wrote the discussion, going over all possible sources of the signals we describe.

(2) We added a quantification of brain motion as Figure S5.

(3) We added an example of blood vessel contraction as Figure 4C.

(4) We added data on the fraction of responsive neurons when measured with GCaMP as Figures 3D-3F.

(5) We added example imaging sites from all imaged regions as Figure S1.

(6) We added GFP response heatmaps of all neurons as Figure S2.

(7) We add a quantification of the relationship between GFP response amplitude and expression level Figure S4.

A detailed point-by-point response to all reviewer concerns is provided below.

**Public Reviews:**

**Reviewer #1 (Public Review):**
Fluorescence imaging has become an increasingly popular technique for monitoring neuronal activity and neurotransmitter concentrations in the living brain. However, factors such as brain motion and changes in blood flow and oxygenation can introduce significant artifacts, particularly when activity-dependent signals are small. Yogesh et al. quantified these effects using GFP, an activity-independent marker, under two-photon and wide-field imaging conditions in awake behaving mice. They report significant GFP responses across various brain regions, layers, and behavioral contexts, with magnitudes comparable to those of commonly used activity sensors. These data highlight the need for robust control strategies and careful interpretation of fluorescence functional imaging data.Strengths:The effect of hemodynamic occlusion in two-photon imaging has been previously demonstrated in sparsely labeled neurons in V1 of anesthetized animals (see Shen and Kara et al., Nature Methods, 2012). The present study builds on these findings by imaging a substantially larger population of neurons in awake, behaving mice across multiple cortical regions, layers, and stimulus conditions. The experiments are extensive, the statistical analyses are rigorous, and the results convincingly demonstrate significant GFP responses that must be accounted for in functional imaging experiments. However, whether these GFP responses are driven by hemodynamic occlusion remains less clear, given the complexities associated with awake imaging and GFP's properties (see below).Weaknesses:(1) The authors primarily attribute the observed GFP responses to hemodynamic occlusion. While this explanation is plausible, other factors may also contribute to the observed signals. These include uncompensated brain movement (e.g., axial-direction movements), leakage of visual stimulation light into the microscope, and GFP's sensitivity to changes in intracellular pH (see e.g., Kneen and Verkman, 1998, Biophysical Journal). Although the correlation between GFP signals and blood vessel diameters supports a hemodynamic contribution, it does not rule out significant contributions from these (or other) factors. Consequently, whether GFP fluorescence can reliably quantify hemodynamic occlusion in two-photon microscopy remains uncertain.

We concur; our data do not conclusively prove that the effect is only driven by hemodynamic occlusion. We have attempted to make this clearer in the text throughout the manuscript. In particular we have restructured the discussion to focus on this point. Regarding the specific alternatives the reviewer mentions here:

a) Uncompensated brain motion. While this can certainly contribute, we think the effect is negligible in our interpretation for the following reasons. First, just to point out the obvious, as with all two-photon data we acquire in the lab, we only keep data with no visible z-motion (axial). Second, and more importantly, uncompensated brain motion results in a net decrease of fluorescence. As regions of interest (ROI) are selected to be centered on neurons (as opposed to be randomly selected, or next to, or above or below), movement will – on average – result in a decrease in fluorescence, as neurons are moved out of the ROIs. In the early days of awake two-photon imaging (when preps were still less stable) – we used this movement onset decrease in fluorescence as a sign that running onsets were selected correctly (i.e. with low variance). See e.g. the dip in the running onset trace at time zero in figure 3A of (Keller et al., 2012). Third, we find no evidence for any brain motion in the case of visual stimulation, while the GFP responses during locomotion and visual stimulation are of similar magnitude. We have added a quantification of brain motion (Figure S5) and a discussion of this point to the manuscript.

b) Leakage of stimulation light. First, all light sources in the experimental room (the projector used for the mouse VR, the optogenetic stimulation light, as well as the computer monitors used to operate the microscope) are synchronized to the turnaround times of the resonant scanner of the two-photon microscope. Thus, light sources in the room are turned off for each line scan of the resonant scanner and turned on in the turnaround period. With a 12kHz scanner this results in a light cycle of 24 kHz (see Leinweber et al., 2014 for details). While the system is not perfect, we can occasionally get detectable light leak responses at the image edges (in the resonant axis as a result of the exponential off kinetics of many LEDs & lasers), these are typically 2 orders of magnitude smaller than what one would get without synchronizing, and far smaller than a single digit percentage change in GFP responses, and only detectable at the image edges. Second, while in visual cortex, dark running onsets are different from running onsets with the VR turned on (Figures 5A and B), they are indistinguishable in ACC (Figure 5C). Thus, stimulation light artefacts we can rule out.

c) GFP’s sensitivity to changes in pH. Activity results in a decrease in neuronal intracellular pH (https://pubmed.ncbi.nlm.nih.gov/14506304/, https://pubmed.ncbi.nlm.nih.gov/24312004/) – decreasing pH decreases GFP fluorescence (https://pubmed.ncbi.nlm.nih.gov/9512054/).

To reiterate, we don’t think hemodynamic occlusion is the only possible source to the effects we observe, but we do think it is most likely the largest.

(2) Regardless of the underlying mechanisms driving the GFP responses, these activity-independent signals must be accounted for in functional imaging experiments. However, the present manuscript does not explore potential strategies to mitigate these effects. Exploring and demonstrating even partial mitigation strategies could have significant implications for the field.

We concur – however, in brief, we think the only viable mitigation strategy (we are capable of), is to repeat functional imaging with GFP imaging. To unpack this: There have been numerous efforts to mitigate these hemodynamic effects using isosbestic illumination. When we started to use such strategies in the lab for widefield imaging, we thought we would calibrate the isosbestic correction using GFP recordings. The idea was that if performed correctly, an isosbestic response should look like a GFP response. Try as we may, we could not get the isosbestic responses to look like a GFP response. We suspect this is a result of the fact that none of the light sources we used were perfectly match to the isosbestic wavelength the GCaMP variants we used (not for a lack of trying, but neither lasers nor LEDs were available for purchase with exact wavelength matches). Complicating this was then also the fact that the similarity (or dissimilarity) between isosbestic and GFP responses was a function of brain region. Importantly however, just because we could not successfully apply isosbestic corrections, of course does not mean it cannot be done. Hence for the widefield experiments we then resorted to mitigating the problem by repeating the key experiments using GFP imaging (see e.g. (Heindorf and Keller, 2024)). Note, others have also argued that the best way to correct for hemodynamic artefacts is a GFP recording based correction (Valley et al., 2019). A second strategy we tried was using a second fluorophore (i.e. a red marker) in tandem with a GCaMP sensor. The problem here is that the absorption of the two differs markedly by blood and once again a correction of the GCaMP signal using the red channel was questionable at best. Thus, we think the only viable mitigation strategy we have found is GFP recordings and testing whether the postulated effects seen with calcium indicators are also present in GFP responses. This work is our attempt at a post-hoc mitigation of the problem of our own previous two-photon imaging studies.

(3) Several methodology details are missing from the Methods section. These include: (a) signal extraction methods for two-photon imaging data (b) neuropil subtraction methods (whether they are performed and, if so, how) (c) methods used to prevent visual stimulation light from being detected by the two-photon imaging system (d) methods to measure blood vessel diameter/area in each frame. The authors should provide more details in their revision.

Please excuse, this was an oversight. All details have been added to the methods.

**Reviewer #2 (Public Review):**
In this study, Yogesh et al. aimed at characterizing hemodynamic occlusion in two photon imaging, where its effects on signal fluctuations are underappreciated compared to that in wide field imaging and fiber photometry. The authors used activity-independent GFP fluorescence, GCaMP and GRAB sensors for various neuromodulators in two-photon and widefield imaging during a visuomotor context to evaluate the extent of hemodynamic occlusion in V1 and ACC. They found that the GFP responses were comparable in amplitude to smaller GCaMP responses, though exhibiting context-, cortical region-, and depth-specific effects. After quantifying blood vessel diameter change and surrounding GFP responses, they argued that GFP responses were highly correlated with changes in local blood vessel size. Furthermore, when imaging with GRAB sensors for different neuromodulators, they found that sensors with lower dynamic ranges such as GRAB-DA1m, GRAB5HT1.0, and GRAB-NE1m exhibited responses most likely masked by the hemodynamic occlusion, while a sensor with larger SNR, GRAB-ACh3.0, showed much more distinguishable responses from blood vessel change.StrengthsThis work is of broad interest to two photon imaging users and GRAB developers and users. It thoroughly quantifies the hemodynamic driven GFP response and compares it to previously published GCaMP data in a similar context, and illustrates the contribution of hemodynamic occlusion to GFP and GRAB responses by characterizing the local blood vessel diameter and fluorescence change. These findings provide important considerations for the imaging community and a sobering look at the utility of these sensors for cortical imaging.Importantly, they draw clear distinctions between the temporal dynamics and amplitude of hemodynamic artifacts across cortical regions and layers. Moreover, they show context dependent (Dark versus during visual stimuli) effects on locomotion and optogenetic light-triggered hemodynamic signals.Most of the first generation neuromodulator GRAB sensors showed relatively small responses, comparable to blood vessel changes in two photon imaging, which emphasizes a need for improved the dynamic range and response magnitude for future sensors and encourages the sensor users to consider removing hemodynamic artifacts when analyzing GRAB imaging data.Weaknesses(1) The largest weakness of the paper is that, while they convincingly quantify hemodynamic artifacts across a range of conditions, they do not quantify any methods of correcting for them. The utility of the paper could have been greatly enhanced had they tested hemodynamic correction methods (e.g. from Ocana-Santero et al., 2024) and applied them to their datasets. This would serve both to verify their findings-proving that hemodynamic correction removes the hemodynamic signal-and to act as a guide to the field for how to address the problem they highlight.

See also our response to reviewer 1 comment 2.

In the Ocana-Santero et al., 2024 paper they also first use GFP recordings to identify the problem. The mitigation strategy they then propose, and use, is to image a second fluorophore that emits at a different wavelength concurrently with the functional indicator. The authors then simply subtract (we think – the paper states “divisive”, but the data shown are more consistent with “subtractive” correction) the two signals to correct for hemodynamics. However, the paper does not demonstrate that the hemodynamic signals in the red channel match those in the green channel. The evidence presented that this works is at best anecdotal. In our hands this does not work (meaning the red channel does not match GFP recordings), we suspect this is a combination of crosstalk from the simultaneously recorded functional channel and the fact that hemodynamic absorption is strongly wavelength specific, or something we are doing wrong. Either way, we cannot contribute to this in the form of mitigation strategy.

Given that the GFP responses are a function of brain area and cortical depth – it is not a stretch to postulate that they also depend on genetic cell type labelled. Thus, any GFP calibration used for correction will need to be repeated for each cell type and brain area. Once experiments are repeated using GFP (the strategy we advocate for – we don’t think there is a simpler way to do this), the “correction” is just a subtraction (or a visual comparison).

(2) The paper attributes the source of 'hemodynamic occlusion' primarily to blood vessel dilation, but leaves unanswered how much may be due to shifts in blood oxygenation. Figure 4 directly addresses the question of how much of the signal can be attributed to occlusion by measuring the blood vessel dilation, but notably fails to reproduce any of the positive transients associated with locomotion in Figure 2. Thus, an investigation into or at least a discussion of what other factors (movement? Hb oxygenation?) may drive these distinct signals would be helpful.

See also our response to reviewer 1 comment 1.

We have added to Figure 4 an example of a positive transient. At running onset, superficial blood vessels in cortex tend to constrict and hence result in positive transients.

We now also mention changes in blood oxygenation as a potential source of hemodynamic occlusion. And just to be clear, blood oxygenation (or flow) changes in absence of any fluorophore, do not lead to a two-photon signal. Just in case the reviewer was concerned about intrinsic signals – these are not detectable in two photon imaging.

(3) Along these lines, the authors carefully quantified the correlation between local blood vessel diameter and GFP response (or neuropil fluorescence vs blood vessel fluorescence with GRAB sensors). To what extent does this effect depend on proximity to the vessels? Do GFP/ GRAB responses decorrelate from blood vessel activity in neurons further from vessels (refer to Figure 5A and B in Neyhart et al., Cell Reports 2024)?

We indeed thought about quantifying this, but to do this properly would require having a 3d reconstruction of the blood vessel plexus above (with respect to the optical axis) the neuron of interest, as well as some knowledge of how each vessel dilates as a function of stimulus. The prime effect is likely from blood vessels that are in the 45 degrees illumination cone above the neuron (Author response image 2). Lateral proximity to a blood vessel is likely only of secondary relevance. Thus, performing such a measurement is impractical and of little benefit for others.

**Author response image 2. sa4fig2:** A schematic representation of the cone of illumination.

While imaging a neuron (the spot on the imaging plane at the focus of the cone of illumination), the relevant blood vessels that primarily contribute to hemodynamic occlusion are those in the cone of illumination between the neuron and the objective lens. Blood vessels visible in the imaging plane (indicated by gray arrows), do not directly contribute to hemodynamic occlusion. Any distance dependence of hemodynamic occlusion in the observed response of a neuron to these blood vessels in the imaging plane is at best incidental.

(4) Raw traces are shown in Figure 2 but we are never presented with the unaveraged data for locomotion of stimulus presentation times, which limits the reader's ability to independently assess variability in the data. Inclusion of heatmaps comparing event aligned GFP to GCaMP6f may be of value to the reader.

We fear we are not sure what the reviewer means by “the unaveraged data for locomotion of stimulus presentation times”. We suspect this should read “locomotion or stimulus…”. We have added heat maps of the responses of all neurons of the data shown in Figure 1 – as Figure S2.

(5) More detailed analysis of differences between the kinds of dynamics observed in GFP vs GCaMP6f expressing neurons could aid in identifying artifacts in otherwise clean data. The example neurons in Figure 2A hint at this as each display unique waveforms and the question of whether certain properties of their dynamics can reveal the hemodynamic rather than indicator driven nature of the signal is left open. Eg. do the decay rate and rise times differ significantly from GCaMP6f signals?

The most informative distinction we have found is differences in peak responses (Figure 2B). Decay and rise time measurements critically depend on the identification of “events”. As a function of how selective one is with what one calls an event (e.g. easy in example 1 of Figure 2 – but more difficult in examples 2 and 3), one gets very different estimates of rise and decay times. Due to the fact that peak amplitudes are lower in GFP responses – rise and decay times will be either slower or noisier (depending on where the threshold for event detection is set).

(6) The authors suggest that signal to noise ratio of an indicator likely affects the ability to separate hemodynamic response from the underlying fluorescence signal. Does the degree of background fluorescence affect the size of the artifact? If there was variation in background and overall expression level in the data this could potentially be used to answer this question. Could lower (or higher!) expression levels increase the effects of hemodynamic occlusion?

There may be a misunderstanding (i.e. we might be misunderstanding the reviewer’s argument here). Our statement from the manuscript that the signal to noise ratio of an indicator matters is based on the simple consideration that hemodynamic occlusion is in the range of 0 to 2 % ΔF/F. The larger the dynamic range of the indicator, the less of a problem 2% ΔF/F are. Imagine an indicator with average responses in the 100’s of % ΔF/F - then this would be a non-problem. For indicators with a dynamic range less than 1%, a 2% artifact is a problem.

Regarding “background” fluorescence, we are not sure what is meant here. In case the reviewer means fluorescence that comes from indicator molecules in processes (as opposed to soma) that are typically ignored (or classified as neuropil) – we are not sure how this would help. The occlusion effects are identical for both somatic and axonal or dendritic GFP (the source of the GFP fluorescence is not relevant for the occlusion effect). In case the reviewer means “baseline” fluorescence – above a noise threshold ΔF/F_0_ should be constant independent of F_0_ (i.e. baseline fluorescence). This also holds in the data, see Figure S4. We might be stating the trivial - the normalization of fluorescence activity as ΔF/F_0_ has the effect that the “occluder" effect is constant for all values of all F_0_.

(7) The choice of the phrase 'hemodynamic occlusion' may cause some confusion as the authors address both positive and negative responses in the GFP expressing neurons, and there may be additional contributions from changes in blood oxygenation state.

Regarding the potential confusion with regards to terminology, occlusion can decrease or increase.

Only under the (incorrect) assumption that occlusion is zero at baseline would this be confusing – no? If the reviewer has a suggestion for a different term, we’d be open to changing it.

Regarding blood oxygenation – this is absolutely correct, we did not explicitly point this out in the previous version of the manuscript. Occlusion changes are driven by a combination of changes to volume and “opacity” of the blood. Oxygenation changes would be in the second category. We have clarified this in the manuscript.

(8) The choice of ACC as the frontal region provides a substantial contrast in location, brain movement, and vascular architecture as compared to V1. As the authors note, ACC is close to the superior sagittal sinus and thus is the region where the largest vascular effects are likely to occur. The reader is left to wonder how much of the ROI may or may not have included vasculature in the ACC vs V1 recordings as the only images of the recording sites provided are for V1. We are left unable to conclude whether the differences observed between these regions are due to the presence of visible vasculature, capillary blood flow or differences in neurovasculature coupling between regions. A less medial portion of M2 may have been a more appropriate comparison. At least, inclusion of more example imaging fields for ACC in the supplementary figures would be of value.

Both the choice of V1 and ACC were simply driven by previous experiments we had already done in these areas with calcium indicators. And we agree, the relevant axis is likely distance from midline, not AP – i.e. RSC and ACC are likely more similar, and V1 and lateral M2 more similar. We have made this point explicitly in the manuscript and have added sample fields of view as Figure S1.

(9) In Figure 3, How do the proportions of responsive GFP neurons compare to GCaMP6f neurons?

We have added the data for GCaMP responses.

(10) How is variance explained calculated in Figure 4? Is this from a linear model and R^2 value? Is this variance estimate for separate predictors by using single variable models? The methods should describe the construction of the model including the design matrix and how the model was fit and if and how cross validation was run.

This is simply a linear model (i.e. R^2) – we have added this to the methods.

(11) Cortical depth is coarsely defined as L2/3 or L5, without numerical ranges in depth from pia.

Layer 2/3 imaging was done at a depth of 100-250 μm from pia, and the same for layer 5 was 400-600 μm. This has been added to the methods.

Overall Assessment:This paper is an important contribution to our understanding of how hemodynamic artifacts may corrupt GRAB and calcium imaging, even in two-photon imaging modes. Certain useful control experiments, such as intrinsic optical imaging in the same paradigms, were not reported, nor were any hemodynamic correction methods investigated. Thus, this limits both mechanistic conclusions and the overall utility with respect to immediate applications by end users. Nevertheless, the paper is of significant importance to anyone conducting two-photon or widefield imaging with calcium and GRAB sensors and deserves the attention of the broader neuroscience and in-vivo imaging community.
**Reviewer #3 (Public review):**
In this study, the authors aimed to investigate if hemodynamic occlusion contributes to fluorescent signals measured with two-photon microscopy. For this, they image the activity-independent fluorophore GFP in 2 different cortical areas, at different cortical depths and in different behavioral conditions. They compare the evoked fluorescent signals with those obtained with calcium sensors and neuromodulator sensors and evaluate their relationship to vessel diameter as a readout of blood flow.They find that GFP fluorescence transients are comparable to GCaMP6f stimuli-evoked signals in amplitude, although they are generally smaller. Yet, they are significant even at the single neuronal level. They show that GFP fluorescence transients resemble those measured with the dopamine sensor GRABDA1m and the serotonin sensor GRAB-5HT1.0 in amplitude an nature, suggesting that signals with these sensors are dominated by hemodynamic occlusion. Moreover, the authors perform similar experiments with wide-field microscopy which reveals the similarity between the two methods in generating the hemodynamic signals. Together the evidence presented calls for the development and use of high dynamic range sensors to avoid measuring signals that have another origin from the one intended to measure. In the meantime, the evidence highlights the need to control for those artifacts such as with the parallel use of activity independent fluorophores.Strengths:- Comprehensive study comparing different cortical regions in diverse behavioral settings in controlled conditions.- Comparison to the state-of-the-art, i.e. what has been demonstrated with wide-field microscopy.- Comparison to diverse activity-dependent sensors, including the widely used GCaMP.Weaknesses:(1) The kinetics of GCaMP is stereotypic. An analysis/comment on if and how the kinetics of the signals could be used to distinguish the hemodynamic occlusion artefacts from calcium signals would be useful.

We might be misunderstanding what the reviewer means by “the kinetics of GCaMP are stereotypic”. The kinetics are clearly stereotypic if one has isolated single action potential responses in a genetically identified cell type. But data recorded in vivo looks very different, see e.g. example traces in figure 1g of (Keller et al., 2012). And these are selected example traces, the average GCaMP trace looks perhaps more like the three example traces shown in Figure 2 (this is not surprising if the GCaMP signals one records in vivo are a superposition of calcium responses and hemodynamic occlusion). All quantification of kinetics relies on identifying “events”. We cannot identify events in any meaningful way for most of the data (see e.g. examples 2 and 3 in Figure 2). The one feature we can reliably identify as differing between GCaMP and GFP responses is peak response amplitude (as quantified in Figure 2).

(2) Is it possible that motion is affecting the signals in a certain degree? This issue is not made clear.

See also our response to reviewer 1 comment 1. In brief, we have added a quantification of motion artefacts as Figure S5, and argue that motion artefacts could only account for locomotion onset responses (there is no detectable brain motion to visual responses) and would predict a decrease in fluorescence (not an increase).

(3) The causal relationship with blood flow remains open. Hemodynamic occlusion seems a good candidate causing changes in GFP fluorescence, but this remains to be well addressed in further research.

We agree – we have made this clearer in the manuscript.

**Recommendations for the authors:**

**Reviewer #1 (Recommendations for the authors):**
(1) Figure 2A shows three neurons with convincing GFP responses, with amplitudes often exceeding 100%. However, after seeing these data, I actually feel less convinced that these responses are related to hemodynamic occlusion. Blood vessel diameter changes by at most a few percent during behavior -- how could such small changes lead to >100% changes in GFP fluorescence?My guess is that these responses might instead be related to motion artifacts, particularly given the strong correlation between these responses and running speed (Figure 2A). One possible way to test this is by examining a pixelwise map of fluorescence changes (dF/F) during running vs. baseline. If hemodynamic effects are involved, one would likely see a shadow of the involved blood vessels in this map. Conversely, if motion artifacts are the primary factor, the map of dF/F should resemble the spatial gradients of the mean fluorescence image. Examining pixelwise maps of dF/F will likely provide insights regarding the nature of the GFP signals.

The underlying assumption (“blood vessel diameter changes by at most a few percent”) might be incorrect here. (Note also, relevant is likely the cross section, not diameter.) See Figure 4A1 and B1 for quantification of example blood vessel area changes - both example vessels change area by approximately 50%. Also note, example 1 in Figure 2 is an extreme example. The example was chosen to highlight that effects can be large. To try to illustrate that this is not typical however, we also show the distribution of all neurons in Figure 2B and mark all three example cells – example 1 is at the very tail of the distribution.

Regarding the analysis suggested, we have added examples of this for running onset to the manuscript (Figure S7). We have examples in which a blood vessel shadow is clearly visible. More typical however, is a general increase in fluorescence (on running onset) that we think is caused by blood vessels closer to the surface of the brain.

(2) Figure 3A shows strong GFP responses during running, while visuomotor mismatch elicit virtually no GFP-responsive neurons. This finding is puzzling, as visuomotor mismatch has been shown by the same group to activate L2/3 neurons more strongly than running (see Figure 3A, Keller et al., 2012, Neuron). Stronger neuronal activation should, in theory, result in more pronounced hemodynamic effects, and therefore, a higher proportion of GFP-responsive neurons. The absence of GFP responses during visuomotor mismatch raises questions about whether GFP signals are directly linked to hemodynamic occlusion.An alternative explanation is that the strong GFP responses observed during running could instead be driven by motion artifacts, e.g., those associated with the increased head or body movements during running onsets. Such artifacts could explain the observed GFP responses, rather than hemodynamic occlusion.

This might be a misunderstanding. Mismatch responses are primarily observed in mismatch neurons. These are superficial L2/3 neurons (possibly the population that in higher mammals is L2 neurons). The fact that mismatch responses are primarily observed in this superficial population is likely the reason they were discovered using two-photon calcium imaging (which tends to have a bias towards superficial neurons as the image quality is best there), and seen in much fewer neurons when using electrophysiological techniques (Saleem et al., 2013) that are biased to deeper neurons. In response to Reviewer #2, we have now also added a quantification of the fraction of neurons responsive to these stimuli when using GCaMP (Figure 3D-F). The fraction of neurons responsive to visuomotor mismatch is smaller than those responsive on locomotion or to visual stimuli.

Thus, based on “average” responses across all cortical cell types (our L2/3 recordings here are as unbiased across all of L2/3 as possible) the response profiles (strong running onset and visual responses, and weak MM responses) are probably what one would expect in first approximation also in the blood vessel response profile. Complicating this is of course the fact that it is likely some cell type specific activity that contributes most to blood flow changes, not simply average neuronal activity.

See response to public review comment 1 for a discussion of alternative sources, including motion artefacts.

(3) Given the potential confound associated with brain motion, the authors might consider quantifying hemodynamic occlusion effects under more controlled conditions, such as in anesthetized animals, where brain movement is minimal. They could use drifting grating stimuli, which are known to produce wellcharacterized blood vessel and hemodynamic responses in V1. The effects of hemodynamic occlusion can then be quantified by imaging the fluorescence of an activity-independent marker. For maximal robustness, GFP should ideally be avoided, due to its known sensitivity to pH changes, as noted in the public review.

Brain motion is negligible to visual stimuli in the awake mouse as well (Figure S5). This is likely the better control than anesthetized recordings – anesthesia has strong effects on blood pressure, heart rate, breathing, etc. all of which would introduce more confounds.

(4) Regardless of the precise mechanism driving the observed GFP response, these activity-independent signals must be accounted for in functional imaging experiments. This applies not only to experiments using small dynamic range sensors but also to those employing 'high dynamic range' sensors like GCaMP6, which, according to the authors, exhibit responses only ~2-fold greater than those of GFP.In this context, the extensive GFP imaging data are highly valuable, as they could serve as a benchmark for evaluating the effectiveness of correction methods. Ideally, effective correction methods should produce minimal responses when applied to GFP imaging data. With these data at hand, I strongly encourage the authors to explore potential correction methods, as such methods could have far-reaching impact on the field.

As discussed above, we have tested a number of such correction approaches for both widefield and two-photon imaging and could never recover a response profile that resembles the GFP response. The “correction method” we have come to favor, is repeating experiments using GFP (i.e. what we have done here).

(5) Several correction approaches could be considered: for instance, the strong correlation between GFP responses and blood vessel diameter (as shown in Figure 4) could potentially be leveraged to predict and compensate for the activity-independent signals. Alternatively, expressing an activity-independent marker alongside the activity sensor in orthogonal spectral channels could enable simultaneous monitoring and correction of activity-independent signals. Finally, computational procedure to remove common fluctuations, measured from background or 'neuropil' regions (see, e.g., Kerlin et al., 2010, Neuron; Giovannucci et al., 2019, eLife), may help reduce the contamination in cellular ROIs. The authors could try some or all of these methods, and benchmark their effectiveness by assessing, e.g., the number of GFP responsive neurons after correction.

Over the years we have tried many of these approaches. A correction using a second fluorophore of a different color likely fails because blood absorption is strongly wavelength dependent, making it challenging to calibrate the correction factor. Neuropil “correction” on GCaMP data, even with the best implementations, is just a common mode subtraction. The signal in the neuropil – as the name implies is just an average of many axons and dendrites in the vicinity – most of these processes are from nearby neurons making a neuropil response simply an average response of the neurons in some neighborhood. Adding the problem of hemodynamic responses (which on small scales will also influence nearby neurons and neuropil similarly) makes disentangling the two effects impossible (i.e. neuropil subtraction makes the problem worse, not better). However, just because we fail in implementing all of these methods, does not necessarily mean the method is faulty. Hence we have chosen not to comment on any such method, and simply provide the only mitigation strategy that works in our hands – record GFP responses.

(6) Given the potential usefulness of the GFP imaging data, I encourage the authors to share these data in a public repository to facilitate the development of correction methods.

Certainly – all of our data are always published. In the early years of the lab on an FMI repository here https://data.fmi.ch/ - more recently now on Zenodo.

(7) As noted in the public review, several methodology details are missing. Most importantly, I could not find the description in the Methods section explaining how fluorescence signals from individual neurons were extracted from two-photon imaging data. The existing section on 'Extraction of neuronal activity' appears to cover only the wide-field analysis, with details about two-photon analysis seemingly absent.

Please excuse the omission – this has all been added to the methods. In brief, to answer your questions:

Were regions of interest (ROIs) for individual cells identified manually or automatically?

We use a mixture of manual and automatic methods for our two-photon data. Based on a median filtered (spatially) version of the mean fluorescence image, we used a threshold based selection of ROIs. This was then visually inspected and manually corrected where necessary such that ROIs were at least 250 pixels and only labelled clearly identifiable neurons.

Was fluorescence within each ROI calculated by averaging signals across pixels, or were signal de-mixing algorithms (e.g., PCA, ICA, or NMF) applied?

We use the average fluorescence across pixels without any de-mixing algorithms here and in all our two-photon experiments. De-mixing algorithms can introduce a variety of artefacts.

Additionally, did the authors account for and correct the contribution of surrounding neuropil?

No neuropil correction was applied. It would also be difficult to see how this would help. If the model of hemodynamic occlusion is correct, one would expect occlusion effects to change on the length scale of blood vessels (i.e. tens to hundreds of microns). Thus, the effect of occlusion on neuropil and cells should be the similar. Neuropil “correction” is always based on the idea of removing signals that are common to both neuropil and somata, thereby complicating the interpretation of the resulting signal even further.

Without these methodological details, it is difficult to accurately interpret the two-photon signals reported in the manuscript.(8) The rationale for using the average fluorescence of a ROI within the blood vessel as a proxy for blood vessel diameter is not entirely clear to me. The authors should provide a clearer justification for this approach in their revision.

Consider a ROI placed within a blood vessel at the focus of the illumination cone (Author response image 3). Given the axial point-spread-function of two-photon imaging is in the range of 0.5 μm laterally and 3 μm axially (indicated by the bicone), emitted photons from the fluorescent tissue outside of the blood vessel but within the two-photon volume will contribute to change in fluorescence in the ROI. A change in the blood vessel volume, say an increase on dilation, would decrease the amount of emission photons reaching the objective by, one, pushing more of the fluorescent tissue outside of the two-photon volume, and two, by presenting greater hemodynamic occlusion to the photons emitted by the fluorescent tissue immediately below the vessel. Conversely, on vasoconstriction there are more emission photons at the objective.

In line with this argument, as shown in Figure 4A1-A2, B1-B2 and C1-C2, we do find that the change in fluorescence of blood vessel ROI varies inversely with the area of the blood vessel. Of course, change in blood vessel ROI fluorescence is only a proxy for vessel size. Extracting blood vessel boundaries from individual two-photon frames was noisy and proved unreliable in the absence of specific dyes to label the vessel walls. We thus resorted to using blood vessel ROI fluorescence as a proxy for hemodynamic occlusion, and tested how much of the variance in GFP responses is explained by the change in blood vessel ROI response.

We have added an explanation to the manuscript, as suggested.

**Author response image 3. sa4fig3:** Average response of ROIs placed within blood vessels co-vary with hemodynamic occlusion.

(9) I find that the Shen et al., 2012, Nature Methods paper has gone quite far to demonstrate the effect of hemodynamic occlusion in two photon imaging. Therefore, I suggest the authors describe and cite this work not only in the discussion but also in the introduction, where they can highlight the key questions left unanswered by that study and explain how their manuscript aims to address them.

We have added the reference and point to the work in the introduction as suggested.

**Reviewer #3 (Recommendations for the authors):**
I appreciate very much that the study is presented in a very clear manner.A few comments that could clarify it even further:(1) Fig. 1: make clear on legend if it is an average of full FOVs.

The traces shown are the average over ROIs (neurons) – we have clarified in the figure legend as suggested.

(2) Give a more complete definition of hemodynamic occlusion to understand the hypothesis in the relationship between blood vessel dilation and GFP fluorescence (116-119). Maybe, move the phrase from conclusion "Since blood absorbs light, hemodynamic occlusion can affect fluorescence intensity measurements" (219-220).

Very good point – we expanded on the definition in the introduction.

(3) For clarity, mention in the main text the method used to assess how a parameter explains the variance (126-129).

Is implemented.

(4) Discuss the possible relationship of the signals to neuronal activity.

We have added this to the discussion.

(5) Discuss if the measurements could provide any functional insights, whether they could be used to learn something about the brain.

We have added this to the discussion.